# A bony-crested Jurassic dinosaur with evidence of iridescent plumage highlights complexity in early paravian evolution

Dongyu Hu[1], Julia A. Clarke[2], Chad M. Eliason[2,3], Rui Qiu[1], Quanguo Li[4], Matthew D. Shawkey[5], Cuilin Zhao[1], Liliana D'Alba[5], Jinkai Jiang[1] & Xing Xu[6]

The Jurassic Yanliao theropods have offered rare glimpses of the early paravian evolution and particularly of bird origins, but, with the exception of the bizarre scansoriopterygids, they have shown similar skeletal and integumentary morphologies. Here we report a distinctive new Yanliao theropod species bearing prominent lacrimal crests, bony ornaments previously known from more basal theropods. It shows longer arm and leg feathers than *Anchiornis* and tail feathers with asymmetrical vanes forming a tail surface area even larger than that in *Archaeopteryx*. Nanostructures, interpreted as melanosomes, are morphologically similar to organized, platelet-shaped organelles that produce bright iridescent colours in extant birds. The new species indicates the presence of bony ornaments, feather colour and flight-related features consistent with proposed rapid character evolution and significant diversity in signalling and locomotor strategies near bird origins.

[1] Shenyang Normal University, Paleontological Museum of Liaoning, Key Laboratory for Evolution of Past Life in Northeast Asia, Ministry of Land and Resources, 253 North Huanghe Street, 110034 Shenyang, China. [2] Department of Geological Sciences and Integrated Bioscience, University of Texas at Austin, Austin, TX 78712, USA. [3] Field Museum of Natural History, Integrative Research Center, 1400S Lake Shore Drive, Chicago, IL 60605, USA. [4] State Key Laboratory of Biogeology and Environmental Geology, China University of Geosciences, 100083 Beijing, China. [5] Department of Biology, University of Ghent, Evolution and Optics of Nanostructures Group, Ledeganckstraat 35, 9000 Ghent, Belgium. [6] Key Laboratory of Vertebrate Evolution and Human Origins of Chinese Academy of Sciences, Institute of Vertebrate Paleontology and Paleoanthropology, Chinese Academy of Sciences, 142 Xiwai Street, 100044 Beijing, China. Correspondence and requests for materials should be addressed to X.X. (email: xingxu@vip.sina.com) or to J.A.C. (email: julia_clarke@jsg.utexas.edu) or to D.H. (email: hudongyu@synu.edu.cn)

Theropod dinosaurs from the Middle-Late Jurassic Yanliao Biota have been among the most significant discoveries informing the bird origins[1–8]. However, with the exception of the bizarre scansoriopterygids, these species have shown some marked skeletal similarities to the slightly younger 'urvogel' *Archaeopteryx*, for example, in skull shape and forelimb proportions[4,5].

The newly discovered Yanliao specimen described here is another exception to the general pattern. This new theropod shows an array of bony features, as well as plumage characteristics and putative melanosome morphologies not previously seen in other Paraves, and thus further informs the pattern of character acquisition close to the origin of avian flight.

## Results

### Systematic palaeontology

Theropoda Marsh, 1881
Maniraptora Gauthier, 1986
Paraves Sereno, 1998
*Caihong juji* gen. et sp. nov.

**Etymology**. *Caihong* is from the Mandarin 'Caihong' (rainbow), referring to the beautiful preservation of the holotype specimen of the animal and the array of insights it offers into paravian evolution; *juji* is from the Mandarin 'ju' (big) and 'ji' (crest), referring to the animal's prominent lacrimal crests.

**Holotype**. PMoL-B00175 (Paleontological Museum of Liaoning), a nearly complete skeleton with associated plumage preserved on a slab and counter slab from Gangou, Qinglong, northern Hebei Province, Tiaojishan Formation, early Late Jurassic, ~161 Myr[9] (Supplementary Figs. 1–5; Supplementary Notes 1 and 2).

**Diagnosis**. A small theropod with the following autapomorphies within Paraves: accessory fenestra posteroventral to promaxillary fenestra, lacrimal with prominent dorsolaterally oriented crests, robust dentary with anterior tip dorsoventrally deeper than its midsection and short ilium (<50% of the femoral length, compared to considerably >50% in other theropods).

**Differentia**. Besides the aforementioned autapomorphies, *Caihong juji* further differs from the Yanliao theropods *Anchiornis huxleyi*, *Xiaotingia zhengi*, *Eosinopteryx brevipenna* and *Aurornis xui* in possessing the following features: a shallow skull with long snout (about 60% of skull length), antorbital fenestra much longer anteroposteriorly than high dorsoventrally, postorbital with extremely short squamosal process and exceedingly long jugal process, caudal vertebral series short (ie, caudal vertebral series/femur length ratio about 2.5, compared to ~4.0 in *Anchiornis huxleyi* and *Aurornis xui*; this ratio has been suggested to be 2.7 in *Eosinopteryx brevipenna*, but the illustrated posterior-most preserved caudal vertebrae are only slightly shorter than the mid-series caudals[5] and thus are not the terminal caudals in that taxon), forelimb proportionally short (about 60% the hindlimb length, compared to about 85% in *Anchiornis huxleyi*, about 80% in *Aurornis xui* and 75% in *Eosinopteryx brevipenna*), forearm proportionally long (ulna and radius longer than humerus, a feature only known in relatively derived birds and in the scansoriopterygid *Yi qi* among theropods) and manual unguals proportionally small (eg, Manual phalanx III-3/III-2 length ratio about 0.5, compared to ~0.9 in *Anchiornis huxleyi* and other Tiaojishan theropods).

*Caihong juji* further differs from *Anchiornis huxleyi*, *Eosinopteryx brevipenna* and *Aurornis xui* but not *Xiaotingia zhengi* in the following features: jugal with long and shallow quadratojugal process, ilium with a long preacetabular process (about 60% of iliac length compared to about 50% in *Anchiornis huxleyi*, *Aurornis xui* and *Eosinopteryx brevipenna*) and comparatively short lower segments of hindlimb (eg, tibiotarsus <120% of femoral length, compared to about 160% in *Anchiornis huxleyi* and about 140% in *Aurornis xui* and *Eosinopteryx brevipenna*).

*Caihong juji* further differs *Xiaotingia zhengi* in the following features: the posterior maxillary ramus shallow (ie, a depth at mid-length considerably smaller than the corresponding portion of the dentary), the surangular with less extensive lateral exposure, a moderately large surangular foramen, the tooth crowns recurved and slender in lateral view, metacarpal IV less robust than metacarpals II and III; and manual phalanx III-2 shorter than metacarpal III (we identify the three manual digits of maniraptorans as II-III-IV following some recent studies[7,10], rather than as I-II-III as in many other studies[11]), ischium with rectangular obturator process (only known in *Anchiornis huxleyi*, *Aurornis xui*, and *Eosinopteryx brevipenna* and *Archaeopteryx* among coelurosaurs), among others. *Caihong juji* further differs *Eosinopteryx brevipenna* in the following features: ilium with relatively deep postacetabular process and extensive tail feathering.

*Caihong juji* further differs *Aurornis xui* in the following features: manual phalanx II-1 about the same in thickness as the radius, ilium with a convex dorsal margin, ischium with posteriorly curved distal end (compared to the unusually anteroventrally curved ischial distal end in *Aurornis*), and extremely short metatarsal I (<15% of the metatarsal length, compared to about 30% in *Aurornis xui*), among others. *Caihong juji* differs from *Pedopenna daohugouensis* in the following features: a relatively robust pedal phalanx I-1 (a length/mid-shaft-diameter ratio of about 4.0, compared to 7.2 in *Pedopenna*), an extremely short metatarsal I (<15% of the metatarsal III length, compared to about 25% in *Pedopenna*), pedal phalanx III-3 considerably longer than III-2 (III-3 slightly shorter than III-2 in *Pedopenna*) and extensive feathering of the pedal digits.

**Description and comparisons**. The holotype is assessed to be an adult based on closure of the neurocentral sutures in all preserved vertebrae. It is small (Supplementary Table 1), estimated to be ~400 mm in total skeletal body length with a body mass of ~475 g[12].

The cranial morphology is well exposed on the left side of the rostrum of the skull and mandible preserved on the counter slab, but poorly exposed on the slab and other parts of the skull and mandible on the counter slab (Fig. 1, Supplementary Fig. 4a, b). In order to understand better the cranial morphology, we prepare the slab from the bottom side, which reveals well most of the left side of the skull and mandible except the rostrum (Supplementary Fig. 4c). In general, the skull and mandible are well preserved, with most elements preserved in their original anatomical positions, though a few elements are slightly displaced (Fig. 1d, e). For example, the nasals are slightly inclined anteroventrally, the left palatine is slightly displaced dorsally, the left lacrimal slightly anteriorly and the braincase slightly anteriorly.

The skull is long (slightly shorter than the femur in length) and shallow (maximum height/length ratio about 0.2). It displays five major openings from lateral view: an elongate-oval-shaped external naris that extends posteriorly beyond the anterior border of the antorbital fossa as in many other basal paravians[4], a hypertrophied elliptical maxillary fenestra that is centrally

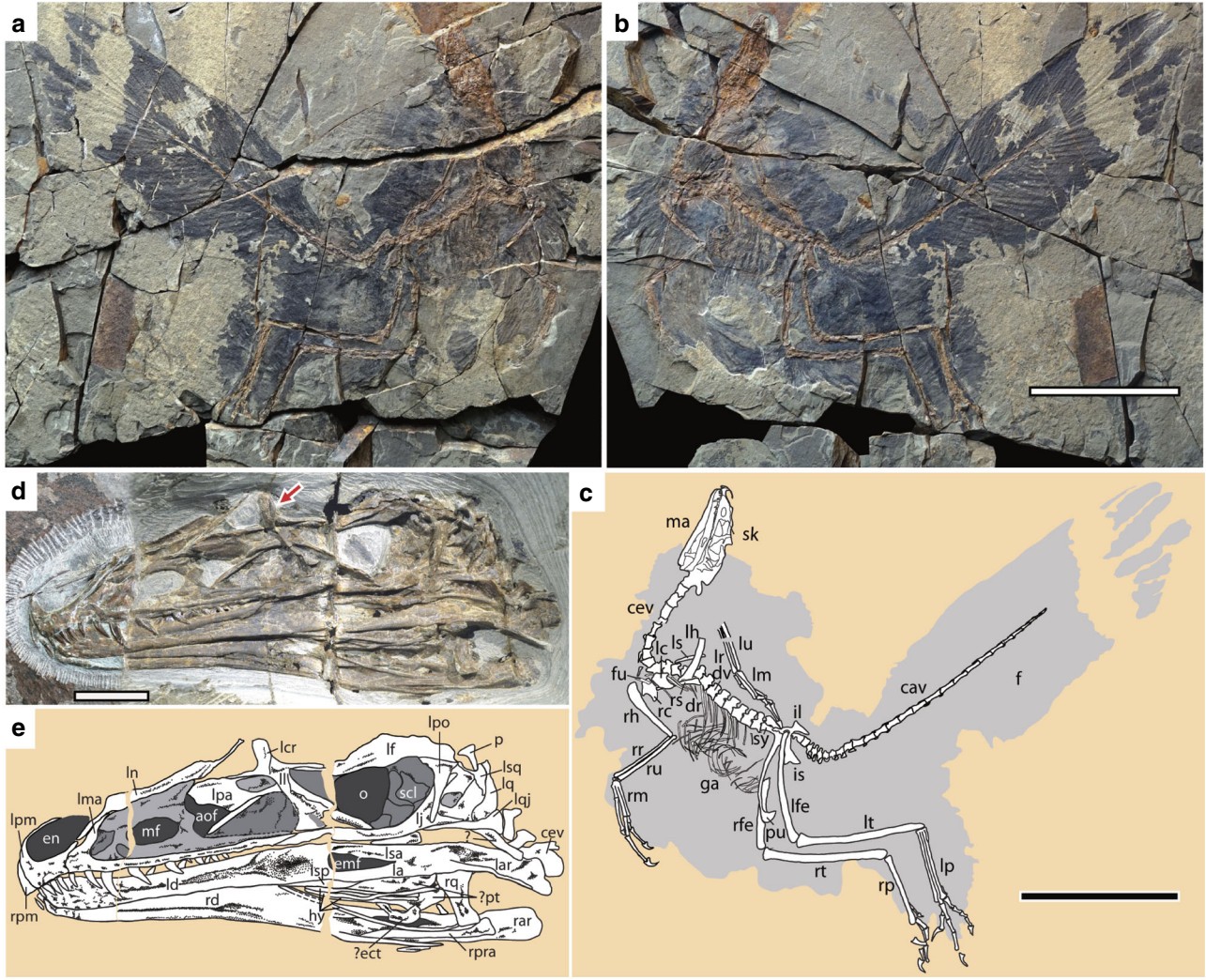

**Fig. 1** *Caihong juji* holotype specimen (PMoL-B00175). Photographs of the slab (**a**) and counter slab (**b**) and line drawing (**c**) of the specimen based on both slabs. Photograph (**d**) and line drawing (**e**) of a composite of the rostrum of the skull and mandible exposed on the counter slab and the post-rostrum cranium exposed on the slab. Arrows indicate lacrimal crests. Question mark indicates uncertain identification. Scale bars: 10 cm **a**–**c**, 1 cm **d** and **e**. aof antorbital fenestra, cav caudal vertebra, cev cervical vertebra, dr dorsal rib, dv dorsal vertebra, ect ectopterygoid, emf external mandibular fenestra, en external naris, f feather, fu furcula, ga gastralia, hy hyoid, il ilium, is ischium, la left angular, lar left articular, lc left coracoid, lcr lacrimal crest, ld left dentary, lf left, frontal, lfe left femur, lh left humerus, lj left jugal, ll left lacrimal, lma left maxilla, lm left manus, ln left nasal, lp left pes, lpa left palatine, lpo left postorbital, lq left quadrate, lqj left quadratojugal, lr left radius, ls left scapula, lsp left splenial, lsa left surangular, lsq left squamosal, lt left tibiotarsus, lu left ulna, ma mandible, mf maxillary fenestra, o orbit, p parietal, pm premaxilla, pt pterygoid, pu pubis, rar right articular, rc right coracoid, rd right dentary, rfe right femur, rh right humerus, rm right manus, rp right pes, rpra right prearticular, rq right quadrate, rr right radius, rs right scapula, rt right tibiotarsus, ru right ulna, scl sclerotic bones, sk skull, sy synsacrum

located, an anteroposteriorly elongated antorbital fenestra that is ~40% of the anteroposterior length of the antorbital fossa and that appears to be extensively medially walled as in the troodontid *Mei*[13], an orbit longer anteroposteriorly than dorsoventrally, and a relatively large infratemporal fenestra.

The premaxilla has a small prenarial portion and a nearly vertical anterior margin more common in dromaeosaurids than in troodontids and basal birds. The subnarial portion is large and longer anteroposteriorly than dorsoventrally. A slender subnarial process contacts a slender nasal subnarial process about the mid-length of the external naris to separate the external naris from the maxilla. The nasal process of the premaxilla is flattened dorsoventrally, a feature seen in troodontids and ornithomimosaurs[14].

The maxilla is relatively shallow element, with the maximum height/length ratio about 0.3. It has a relatively small and high

anterior ramus and a slender subantorbital-fossa ramus that has parallel dorsal and ventral margins and a robust posterior portion. There is a shallow groove along the lateral surface of the subantorbial-fossa ramus in which there are a row of antero-posteriorly-elongate, large foramina. The antorbital fossa contains a large promaxillary fenestra as in some basal paravians including *Archaeopteryx*[4], posteroventral to which is an accessory fenestra. Both fenestrae are at least partially walled medially. The maxillary fenestra is separated from the antorbital fenestra by a narrow interfenestral bar, the posterior margin of which is scalloped in two places, indicating the posterior ends of the narial passage and a ventral canal connecting the antorbital and maxillary fenestrae as in troodontids[15].

The nasal is a major element contributing to the roof of the snout. It sends anteroventrally a slender subnarial process to the level of the mid-length of the external naris, which

overlaps the lateral surface of the subnarial process of the premaxilla.

The lacrimal is tetra-radiated, with an anterior process, a posterior process, a descending process and a fourth dorsolateral process. The anterior process is not well exposed, and thus it is not known whether the anterior process is highly elongated as in troodontids[14]. As in many paravians[16], a posterior process is present, and furthermore, it is long as in many deinonychosaurs[16]. The descending process of the lacrimal has an anteroposteriorly relatively broad lateral surface and expanded ventral end. One striking feature is the presence of a fourth dorsolateral process at the junctional area of the aforementioned three processes, which extends laterally first and then curves dorsally. This process is robust and has a somewhat horned shape in lateral view, unlike the other three processes which are laminal. In many troodontids, the lacrimal extends laterally to form a horizontal shelf over the anterodorsal corner of the orbit[16,17] and in the unenlagiid Austroraptor, a fourth process, which projects posterolaterally, is present lateral to a tiny posterior process[18].

The frontal contributes to the dorsal border of the large orbit. Posteriorly, the frontal is arched. The internal surface of the frontal bears a prominent crista cranium along the orbital edge, which appears to be more prominent posteriorly than anteriorly. The parietal exposes little, but the posterolateral corner of the left parietal is visible. The lateral end of the transverse nuchal crest is not particularly prominent and a short and relatively broad posterolateral process is visible.

The postorbital has an extremely long jugal process, which extends ventrally close to the ventral border of the orbit, and appears to have an extremely short squamosal process and a frontal process that is not strongly upturned. A prominent eminence appears present at the junction with the jugal process, but, this could be a preservation artefact.

The jugal has a very shallow but strap-like suborbital ramus, along the ventral edge of the lateral surface of which is a prominent ridge, a feature also known in some troodontids and basal dromaeosaurids. The postorbital process orients much more posteriorly than dorsally and its great length suggests a long overlapping contact with the jugal process of the postorbital. The quadratojugal process (the posterior process) is long and slender, suggesting the presence of a long contact with the quadratojugal.

The quadratojugal has an extremely long jugal process, which is considerably thickened mediolaterally. The squamosal process appears to be very short, suggesting the absence of the quadratojugal-squamosal contact. There is no distinct posterior process, which is present in dromaeosaurids[19]. The quadrate is strongly curved, and appears to lack a large quadratojugal flange and a large lateral flange. Unusually among paravians, the dorsal end of the quadrate lies more anteriorly than the ventral end, but this might be a preservation artefact. The palatine is a large tetraradiate element. It has a distinctive pneumatic fossa on the ventral surface. Sclerotic ossicles are preserved within the orbit. They are thin plates with a sub-rectangular outline, but more details are not obtainable.

The dentary is a long and shallow bone, roughly triangular in lateral view as in troodontids[14]. The anterior end is not slightly downturned as in Sinovenator[20], and the anterior portion is deep dorsoventrally, even greater in depth than the middle portion. As in troodontids and unenlagiids[14,21], there is a groove on the lateral surface of the dentary, which is narrow anteriorly but much wider posteriorly. Elongated vascular foramina are located within the groove as in troodontids and unenlagiids[14,15,21]. In medial view, a centrally located Meckelian canal is extremely narrow and deep anteriorly, and becomes much wider posteriorly.

The surangular forms more than half of the lateral surface of the posterior portion of the mandible. The anterior process of the surangular bears a groove on the lateral surface for contacting the dentary and it forms the dorsal and posterior border of the external mandibular fenestra. The angular forms the ventral border of the large external mandibular fenestra. The anterior portion of the angular is shallow and the posterior portion deep in lateral view, but it appears not to be as curved as many other deinonychosaurs.

The splenial has a relatively large lateral exposure as in deinonychosaurs[16] and possibly Archaeopteryx[4]. It has a long and shallow posterior process, which contacts the ventral margin of the prearticular and angular, and its anterior portion is not well exposed. Two ceratobranchial elements from the hyoid are preserved.

Caihong resembles basal troodontids and to a lesser degree basal dromaeosaurids in dental features (Supplementary Fig. 4d–h): anterior teeth are slender and closely packed, but middle and posterior teeth are more stout and sparsely spaced. Serrations are absent in the premaxilla and anterior maxilla; the more strongly recurved mid-dentary teeth have posterior serrations, and the posterior-most dentary teeth have short stout crowns with large, apically hooked serrations along both mesial and distal carina. One unusual feature shared with some basal troodontids such as Mei[13] is the extremely long maxillary tooth row: the posterior-most maxillary tooth is located close to the preorbital bar, unlike most maniraptorans that have a short maxillary tooth row terminating considerably anterior to the preorbital bar.

Caihong has probably 10 cervical vertebrae, no more than 13 dorsal, 5 sacral and 26 caudal vertebrae (Figs. 1 and 2). The cervical centra appear to be weakly angled in lateral view as in Archaeopteryx and Anchiornis[2,22]. Slender cervical ribs are parallel to the cervical series and they appear to be long and overlap with neighbouring ribs (Fig. 2a). The dorsal centra are anteroposteriorly long and lack distinctive pneumatic fossae or foramina (Fig. 2b) as in basal deinonychosaurs and Archaeopteryx[20]. The dorsal neural spines are posteriorly expanded distally (Fig. 2b). The sacrum comprises five vertebrae, but little can be said about their morphology.

A complete caudal series comprises 26 vertebrae and it measures only 2.5 times femoral length, proportionally much shorter than those of all other Yanliao taxa[2,6,23]. The proportionally short bony tail is due to the relatively small size of each vertebra. As in other paravians, the anterior caudals are short and with prominent transverse processes and neural spines (Fig. 2c); more posterior ones become longer and bear smaller transverse processes and neural spines (Fig. 2d, e). The transition point, which is determined by the absence of transverse process, occurs between caudal 7 and 8. As in basal paravians[20], the middle and posterior caudal vertebrae are comparatively elongate (ie, the longest caudal vertebra in the midseries is 2.3 times as long as the anterior-most caudal vertebrae). The plate-like chevrons all are short, with the first three longer dorsoventrally than wide anteroposteriorly, the next two sub-equal in length and width, and more posterior ones wider than long, a condition seen in Avialae.

As in basal troodontids and other Tiaojishan theropods, the scapula is slightly shorter than the humerus and the rectangular coracoid has a prominent coracoid tubercle and a relatively small postglenoid process (Fig. 3a). The furcula is relatively slender. Unlike oviraptorosaurs and dromaeosaurids, but as in troodontids, other Tiaojishan theropods, and some basal avialans, ossified sternum and uncinate processes are absent.

The forelimbs in Caihong measure only about 60% the hindlimb length, proportionally shorter than most other basal paravians[2,23]. The humerus is nearly as robust as the femur in most basal paravians (Fig. 3b), but it is proportionally shorter

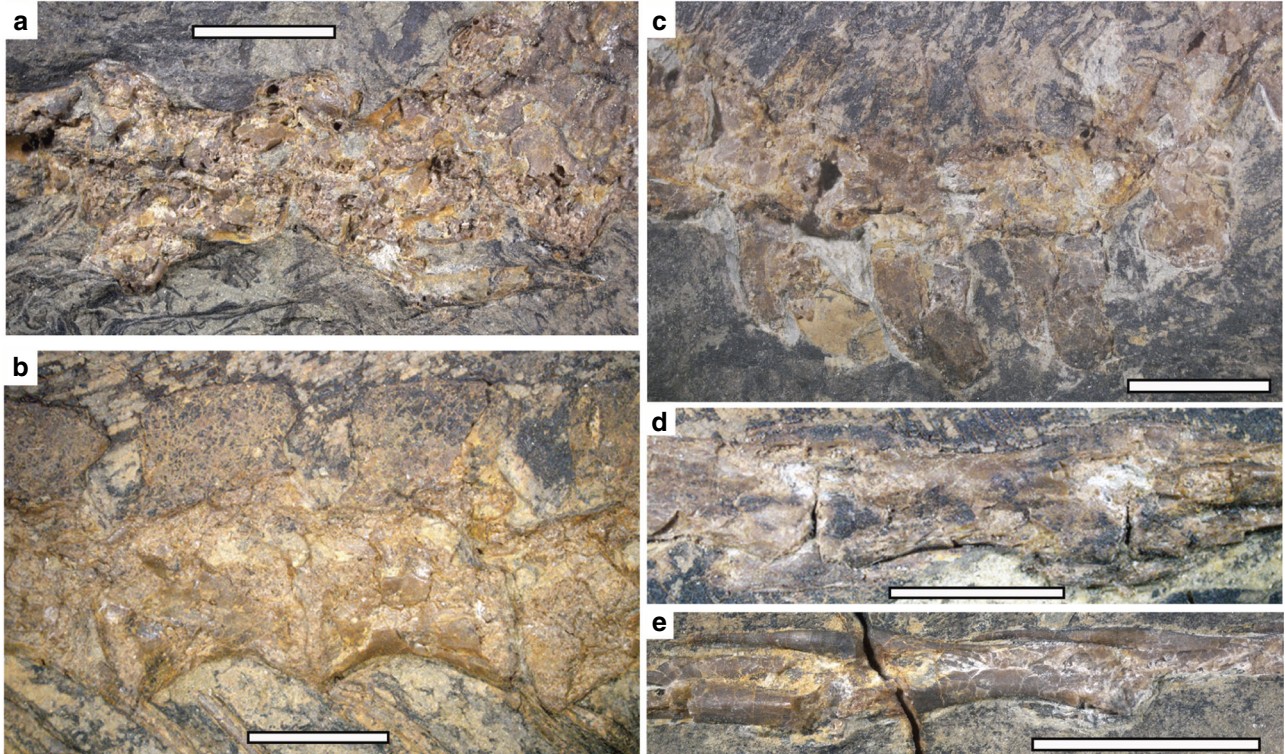

**Fig. 2** Photographs of the vertebral column of *Caihong juji*. **a** Posterior cervical vertebrae in dorsal view; **b** middle dorsal vertebrae in lateral view; **c** anterior caudal vertebrae in dorsal view; **d** middle caudal vertebrae in lateral view; **e** posterior caudal vertebrae in lateral view. Scale bars: 0.5 cm

than in other Tiaojishan theropods and many basal paravians (eg, a humerus/femur length ratio is about 0.6, compared to about 1.0 in *Anchiornis*, 0.8–0.9 in other Tiaojishan theropods)[24,25]. However, the ulna and radius are longer than the humerus, a feature so far known only in flighted avialan taxa among theropods[26]. As in basal troodontids, other Tiaojishan theropods, and to a lesser degree, *Archaeopteryx*[4], the ulna is weakly bowed and the radius is only slightly thinner than the ulna (Fig. 3b, c). The manus resembles those of other Tiaojishan theropods and to a lesser degree basal troodontids: a relatively long metacarpal II (about 40% of the metacarpal IV length), a straight and relatively robust metacarpal IV, a long phalanx II-1 (close in length to metacarpal III), a long phalanx III-2 (about 1.7 times as long as phalanx III-1), and the combined lengths of the subequally long phalanx IV-1 and IV-2 considerably smaller than IV-3 length (Fig. 3b, c; II–IV sensu Xu et al.[10]).

The pelvis closely resembles those of other Tiaojishan theropods and *Archaeopteryx*[4]. The ilium has a preacetabular process considerably longer than the postacetabular process that is slightly downturned; the pubis is slightly curved posteriorly and it has a strongly hooked boot as in the unenlagiids; the ischium is a strap-like element that is extremely short and broad anteroposteriorly, and most unusually, it has a large rectangular obturator process, in striking contrast to nearly all coelurosaurs except *Anchiornis* and *Archaeopteryx* that have a triangular obturator process on the ischium[4] (Fig. 3d).

The hind limbs are highly elongated, measuring about 3.1 times the length of the dorsal series. The sub-arctometatarsalian pes has a hallux that is proportionally smaller than that of most other basal paravians and a moderately enlarged and hyper-extensible second pedal digit (Fig. 3d).

Feathers are well preserved over the whole body, except for the anterior portion of the rostrum and unguals (Fig. 1a–c), but in many cases, they are too densely preserved to display both gross and fine morphological features (eg, number and general outline of feathers and details related to rachis, barbs and barbules). It is noteworthy that the preserved morphologies are not necessarily the true morphologies of these fossil feathers as preservation can alter the morphology of fossil feathers[27].

*Caihong*'s body contour feathers are proportionally longer than those of other known non-avialan theropods[2,5,28]. Feathers attached to the skull and neck are either long (~20 mm in length) and linear, possibly stiff (Fig. 4a; Supplementary Fig. 5) or short (~10 mm) and sinuous in preserved aspect (Fig. 4b). Relatively long feathers (~40 mm) near the chest region and other parts of the body (eg, parts of the forelimb) show sub-parallel, thick and linear barbs (Fig. 4c).

As preserved, the proximal primaries and distal secondaries are among the longest remiges (>100 mm long). These feathers are ~2.4 times as long as the humerus and have narrow rachises (Fig. 4d). While the bony forelimb is shorter than in *Anchiornis*, these remiges are striking more elongated than in that taxon (Fig. 3b). One unusual feature is the preservation of several slender feathers associated with the right pollex (Fig. 4e). These feathers are not completely preserved (missing the distal portions), but the preserved portions suggest that they are pennaceous. Because these feathers are similar in anatomical position to the alula in relatively derived birds, we tentatively identify them as a form of alula, although their function might not be necessarily the same as the alula in relatively derived birds. Nevertheless, similar feathers have been also reported in *Microraptor*, a non-avialan theropod nearly 40 million years younger than *Caihong*[28].

Large pennaceous feathers are preserved posterior to the tibiotarsus and the metatarsus (Figs. 3d, 4f). The longest of these well exceeds any such feathers known in *Eosinopteryx*[5], *Aurornis*[6], and *Archaeopteryx*[29]. These feathers display well-organized vanes on either side of the extremely thin rachis, which

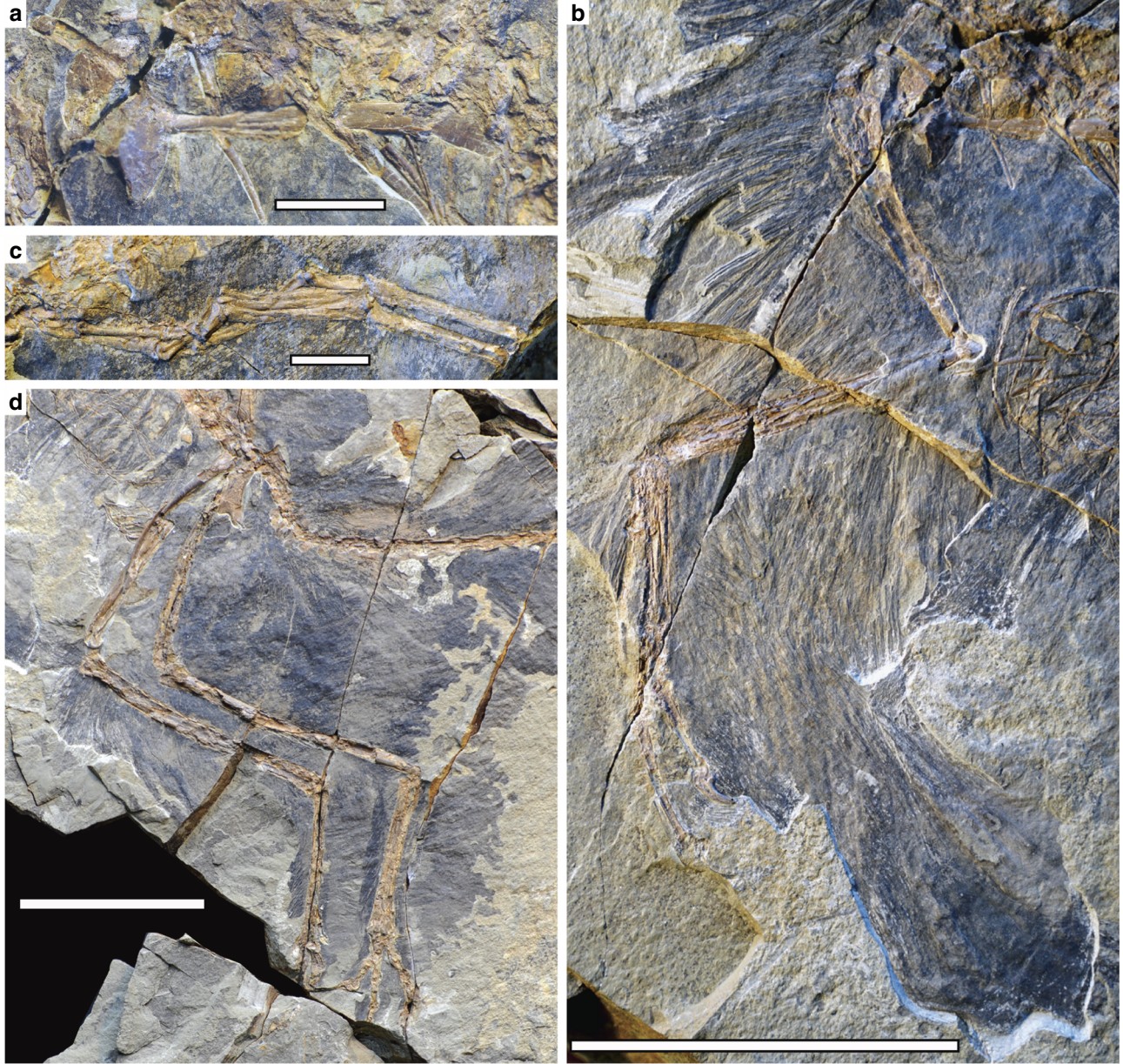

**Fig. 3** Photographs of the pectoral girdle and limbs of *Caihong juji*. **a** Right scapula and coracoid in medial and proximal views; **b** right forelimb; **c** left manus; **d** left and right hind limbs. Scale bar: 1 cm in **a**, **c**, and 5 cm in **b**, **d**

are formed by densely spaced parallel barbs (eg, four barbs in a length of 0.6 mm close to the tip of a feather attaching to the mid-length of the tibiotarsus). In some cases, pennaceous barbules are visible in these feathers (Fig. 4j), which are inferred to be present but have rarely been preserved in other non-avialan theropod fossils[30]. The tibial feathers have barbs with an angle of about 25 degrees to the rachis, forming nearly symmetrical vanes on either side of the rachis.

Small, distally deflected feathers are also preserved anterior to the proximal tibiotarsus, metatarsus, and on the pedal digits (Fig. 3d). The closed vanes are not obvious in most metatarsal and arm feathers. Instead, most preserved metatarsal feathers display densely spaced parallel barbs with nearly same orientation, so are the arm feathers including those that are identified as remiges (Fig. 3b, d). However, several arm feathers show densely spaced parallel barbs orienting in opposite directions on either side of the rachis, suggesting that these feathers are pennaceous (Fig. 4d), and so are several metatarsal feathers.

Hindlimb feathering (Fig. 3d) of *Caihong juji* is seen in *Anchiornis huxleyi*, and other Tiaojishan theropods[2,31]. The new species is particularly similar to the first taxon in having extensively feathered toes[2]. *Eosinopteryx brevipenna* has been suggested to have reduced tail and hindlimb plumages, but specimens of *Anchiornis huxleyi* display variable plumages in terms of not only feather distribution, but also feather size and shape. Some closely related Tiaojishan theropods not differentiated by osteological features may need reassessment of their taxonomic status.

Large pennaceous tail feathers (eg, 112 mm length and 16 mm width in a posterior caudal) attach to all but the most anterior caudal vertebrae, contributing to a robust and broad tail with a sub-rectangular outline (Fig. 1a–c). Feathers attaching to the most anterior caudals are less well-organized and there is no clear evidence for the presence of closed vanes. However, in some cases, closely spaced parallel barbs are visible, which suggest that these feathers may have also been pennaceous. As preserved,

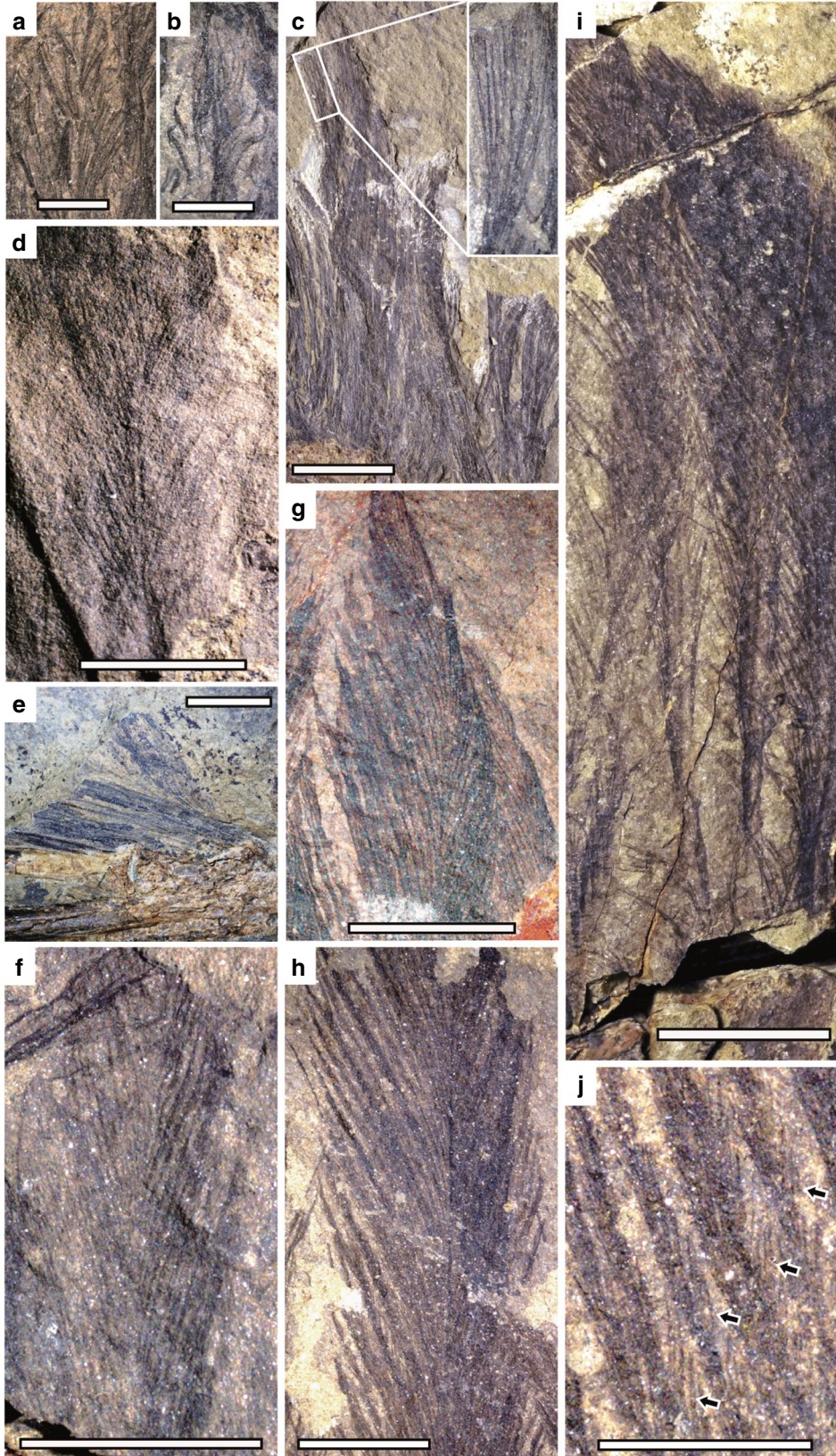

**Fig. 4** Feathers preserved in *Caihong juji*. Feathers attached to the neck: (**a**) pennaceous feathers and (**b**) down-like feather; (**c**) feathers attached to the chest and trunk region; (**d**) primary or secondary remige near the carpal joint; (**e**) alula; (**f**) pennaceous feather on the tibiotarsus; (**g**) anterior rectrix; (**h**) posterior rectrix; (**i**) middle rectrix; (**j**) close-up of barbs and barbules. Scale bars are 0.2 cm in **a**, **b**, **j**; 0.5 cm in **d**, **e**, **f**, **g**, **h**; and 1 cm in **c**, **i**, respectively. Arrows in **j** indicate barbules of a rectrix

feathers attaching to middle and posterior caudal vertebrae are clearly pennaceous, and each caudal vertebra has approximately four associated feathers comprising rectrices and covert feathers. These feathers display well preserved closed vanes (Fig. 4g–i; Supplementary Fig. 5), which are indicated by densely spaced parallel barbs orienting in opposite directions on either side of the rachis that often results in a crossing pattern between the neighbouring feathers. In some cases, barbules are visible in these feathers. For example, some barbules preserved in a middle rectrix are up to 2 mm long and they are parallel to each other and set an acute angle to the barbs (about 15°). These rectrices are among the largest feathers preserved on this specimen. For example, one rectrix attaching to a middle caudal measures about 80 mm in length and 9 mm in maximum width, and one rectrix attaching to the second last caudal measures about 110 mm long and 15 mm wide. Rachises in all these feathers are extremely thin. For example, the rachis of a rectrix attaching to the second last caudal, one of the largest rectrices, measures about 1 mm in diameter about its mid-length. The exceptionally preserved rectrices display proximodistal variation in barb density and angle from fewer higher-angled barbs to more numerous and elongate lower-angled barbs (Fig. 4i). There are four barbs within a length of about 1 mm close to the tip of a feather attaching to a middle caudal, and the same barbs from the leading vane set an angle of about 15 degrees to the rachis and those from the trailing vanes set an angle of about 25°. The posterior rectrices have larger barbs (eg, four barbs within a length of 1.8 mm close to the tip of a rectrix attaching to the second last caudal), which seem to set a larger angle with the rachis (eg, 25° for the leading barbs and 30° for the trailing barbs). Anterior and mid-series rectrices have asymmetrical vanes (eg, mid-series rectrix with the inner vane about twice as wide as the outer vane; Fig. 4g) and attach to the bony tail at a higher angle than the posterior rectrices. Vane asymmetry decreases in the posteriorly directed terminal rectrices, which show nearly symmetrical vanes (Fig. 4h).

In general, the tail feathering of *Caihong* resembles those of *Archaeopteryx*[29] and the troodontid *Jinfengopteryx*[32] in having large rectrices attaching to either side of the caudal series to form a frond-shaped tail (Fig. 1; Supplementary Figs. 3, 5), a unique feature that has been suggested to represent a synapomorphy for the Avialae[33]. *Anchiornis* and other Tiaojishan theropods seem to have a similar tail feathering[25], but their rectrices are smaller and more leaf-like, forming a narrower frond-shaped feathery tail. Furthermore, the available specimens of *Anchiornis* and its close relatives have highly variable tail feathering, with most of them displaying more plumulaceous rather than pennaceous tail feathers and some displaying more pennaceous than plumulaceous feathers. In general, the feathery tail shapes of non-avialan paravian theropods are diverse and some of them appear to have functioned as ornaments[1,28,34–37].

Several other feather morphotypes are also preserved around different regions of the body. One morphotype is represented by a series of barbs joined basally to form a simply branching structure without rachis (Fig. 4b), which is also seen the dromaeosaurid *Sinornithosaurus*[38]. However, some recent studies suggest that this morphotype might represent a derived condition within birds[39] and its presence in non-avialan theropods might be a preservation artefact[27]. A variation of this morphotype is that the barbs are stiff rather than soft, and nearly parallel to each other for most of their length, and also much longer and thicker than the former (Fig. 4a). The other morphotype is represented by a series of long and narrow feathers preserved over the humerus (Fig. 4c). They can be up to at least 45 mm in length but only about 1.5 mm in width. These feathers are composed of an extremely narrow membrane-like leading vane, a rachis that is about 0.3 mm wide (even wider than the leading vane), and a

trailing vane that seems to be open with very long barbs attaching to the rachis at a very small angle.

Given the available information on the plumages of basal paravians[40], it is difficult to draw a conclusion whether rectrices with asymmetrical vanes and/or coverts on the tail represent synapomorphies for the Avialae or a more inclusive clade or independently evolved features in early paravian evolution. Alula seems to be absent in several basal birds such as *Archaeopteryx* and *Confuciuornis*, and thus even if the small feathers associated with the pollex of *Caihong juji* and *Micororaptor* can be confirmed as a form of alula, they must have been independently evolved several times early in paravian evolution given the patchy distribution of this feature among early paravian theropods.

**Nanostructures and inference of plumage colour in the *Caihong juji* holotype.** Scanning electron microscope (SEM) imaging was used to characterize 2460 structures from 66 samples of feathers from across the body (Fig. 5a–d; Supplementary Figs. 6, 7). Cross-sectional focused ion beam (FIB)-SEM images reveal distinct differences between known fossil bacteria and the nanostructures in *Caihong* (Supplementary Figs. 8, 9). SEM data show that the nanostructures recovered from fossil feathers are comparable in size and shape to extant avian melanosomes. Focused ion beam-scanning electron microscopy data show that these structures are solid (ie, uniformly electron-dense; Supplementary Figs. 8, 9). Because bacteria are expected to be electron lucent in the core ('hollow'; see Supplementary Fig. 9 with reproduced Fig. 7c from ref. [41]), these data (the electron-dense cores) as well as absence of any observed evidence of binary fission, do not support a microbial hypothesis for the origin of these nanostructures but do support melanosome identity (Methods). The nanostructures from *Caihong* were compared to an expanded melanosome data set for extant birds (Figs. 5e–h, 6; Methods; Supplementary Tables 2–4; Supplementary Figs. 10–12). Analysis of the structures from *Caihong* indicated that this species occupies a distinct area of morphospace relative to other extinct basal paravians, including *Anchiornis* from the same deposits (Fig. 6a; Supplementary Figs. 12, 13). Rarefaction analyses indicate the structures observed in pinnate feathers of basal paravians are likely to be as diverse as extant bird melanosomes, although this predicted variation is estimated to not yet be fully sampled (Supplementary Figs. 12, 13).

Based on the assumption that these structures are melanosomes, most of the plumage of *Caihong* is predicted by discriminant function analysis as black, with iridescence primarily on the head, chest and, to a lesser extent, the base of the tail (Supplementary Table 4, Supplementary Fig. 6). The morphospace of the platelet-shaped structures overlaps that associated with iridescent colours of vivid, highly variable hues in hummingbirds and swifts (Figs. 5e–g, 6b)[42–44], but also with grey and black colours in penguins[45]. However, unlike melanosomes in those penguin grey or black feathers, the structures in the 21 samples from the fossil appear aligned in sheets and dorsoventrally compressed (Fig. 5; Supplementary Fig. 7). This organization was further confirmed in SEM images taken on a rotating stage at three different angles to the preserved structures (Supplementary Fig. 8).

Other specimens from the same deposits (eg, *Anchiornis*) do not show similar platelet morphologies or organization (Supplementary Fig. 14)[25,34], platelet-shaped nanostructures in *Caihong* occur only in certain body regions (Supplementary Figs. 6, 7), and nanostructures in nearby regions were statistically more similar to each other than more distantly related anatomical areas (Mantel test $r = 0.16$, $p = 0.001$; Supplementary Fig. 15). Thus, flattening of the nanostructures by compaction during burial is considered

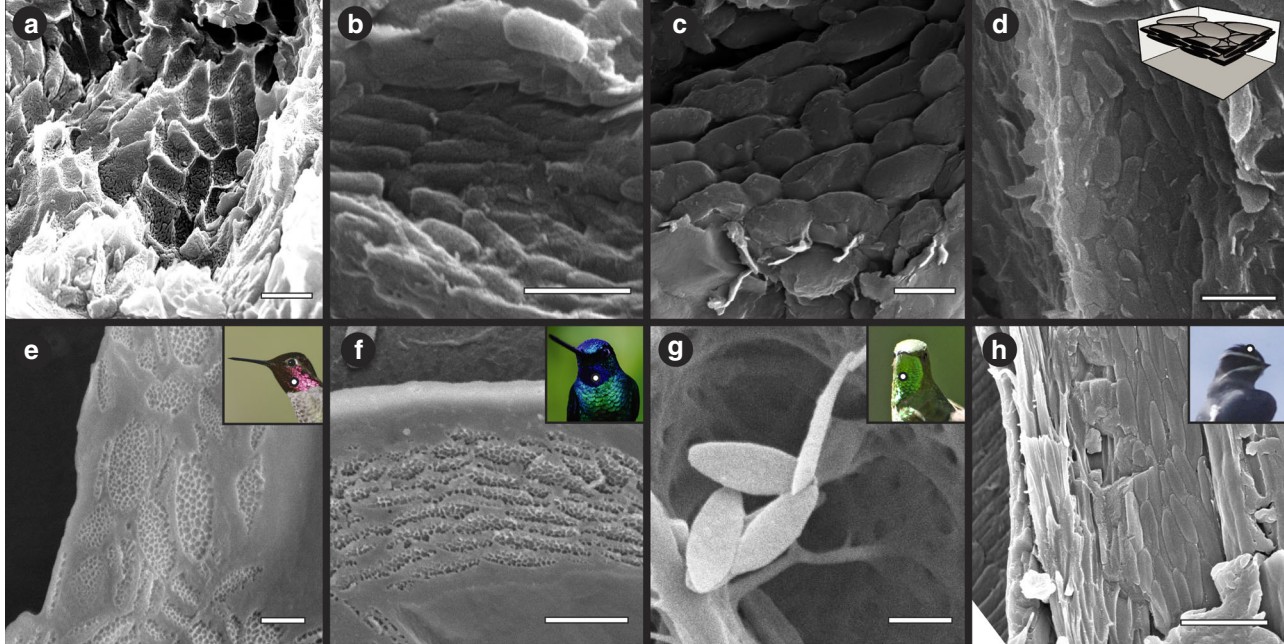

**Fig. 5** Platelet-like nanostructures in *Caihong juji* and melanosomes in iridescent extant feathers. **a–d** Fossilized nanostructures from *Caihong* feathers preserved as molds in a neck feather (**a**) and three-dimensional preservation in a neck feather, with SEM stage rotated 45° to show 3D platelet morphology (**b**), a back feather with SEM stage at 0° (**c**) and a neck feather showing nanostructure packing (**d**); **e** Anna's hummingbird (*Calypte anna*) showing overlapping melanosomes within a feather barbule; **f** white-tailed starfrontlet (*Coeligena phalerata*) showing stacking and interior morphology (air bubbles) of melanosomes in a feather barbule; **g** black-tailed trainbearer (*Lesbia victoriae*) showing exterior surface and morphology of isolated melanosomes in a feather barb; **h** moustached treeswift (*Hemiprocne mystacea*) showing densely packed melanosomes in the barbule of a crown feather. Inset in **d** illustrates 3D stacking of platelet-shaped nanostructures. All scale bars: 1000 nm. Photo credits: Chao PC (**e**, license CC BY-2.0, https://creativecommons.org/licenses/by/2.0/legalcode), Misty Vaughn **f**, Lip Kee (**g**, license CC BY-SA 2.0, https://creativecommons.org/licenses/by-sa/2.0/legalcode), Lip Kee (**h**, CC BY-SA 2.0, https://creativecommons.org/licenses/by-sa/2.0/legalcode)

unlikely. While melanosomes from internal organs can also fossilize, our sampling was limited to feathers, and we did not observe any 'halo' around the body cavity that may be diagnostic of internal melanosome migration[46]. We also modelled the effect of taphonomic shrinkage[47] (Methods). In most cases, our results were not affected by this taphonomic bias (Supplementary Tables 3, 4; Supplementary Fig. 12). If this occurred, the original sizes of these structures in the 21 regions would overlap even more of the iridescent hummingbird melanosome morphospace (Supplementary Figs. 16, 17).

## Discussion

Recovered as a basal deinonychosaur (Fig. 6c; Supplementary Note 3), the Oxfordian *Caihong* has longer arm and leg feathers than its contemporary *Anchiornis* and large feathers forming a tail surface area larger than in the younger *Archaeopteryx*. Furthermore, it shows the earliest asymmetrical feathers and proportionally long forearms in the theropod fossil record. This indicates locomotor differences among these closely related Jurassic paravians and has implications for understanding the evolution of flight-related features. The character combination in *Caihong* is further consistent with significant decoupling of integumentary and musculoskeletal innovation close to the origin of birds. Loss of skeletal ornaments and changes in relative forelimb length are seen to lag behind rapid plumage evolution and the acquisition of an estimated melanin-based colour diversity comparable to living birds (Supplementary Fig. 12). Integumentary structures may largely replace the signalling role of bony features close to the origin of aerial locomotion.

*Caihong* has both prominent bony crests on the skull as well as feathers on the head, chest and tail that are estimated to be iridescent. Basal paravian theropods typically do not possess bony ornaments, which are otherwise widely distributed in dinosaurs[48,49] and have been associated with rapid rates of evolution in more basal theropods[49], but often display ornamental feathers distinct in shape[1,28,35] and colour[7,34,35]. *Caihong* provides the earliest evidence for organized platelet-shaped nanostructures, here interpreted as melanosomes, in dinosaurian feathers (Fig. 5; Supplementary Figs. 7–9). Melanosomes with a similar morphology to the structures observed here have evolved several times in extant birds, and are always associated with iridescence[42].

Tone or precise hue created by light scattering from the platelet structures, if melanosomes, cannot be reconstructed in *Caihong* because that is determined by the precise spacing of the photonic nanostructures in vivo as well as the distribution of keratin and (in rare cases) by the presence of other pigments[50]. Solid platelets are known to produce iridescent colours in songbirds and swifts, while hollow platelets yield these colours in other birds (eg, ibises, trogons, other passerines, hummingbirds)[42]. Although *Caihong*'s platelet-like structures are most similar in shape to hummingbird melanosomes of the included taxa (Figs. 5, 6b), the interior of hummingbird melanosomes is pitted with air bubbles (Fig. 5f)[43] while the fossilized structures show no evidence of hollowness (Supplementary Figs. 8, 9). Indeed, *Caihong*, and the newly-reported extant avian occurrence of solid platelets in the bright, iridescent feathers of trumpeters (Psophiidae) (Figs. 5, 6) may provide further evidence[51,52] for developmental decoupling of melanosome shape and internal morphology (hollow or solid). If other pigments were present (carotenoids, pterins), as in living birds, they would not affect inference of potential structural colour in the new taxon: such pigments are not known to negate structural colour, but may modify the specific tone or hue[50], and

are, in any case, unknown in *Caihong*. Additional non-melanin pigments in the feathers predicted as black would likely be masked by the strong light absorption of melanin[53]. We note, however, that our inferences of colour in *Caihong* and comparisons with other feathered dinosaurs with these structures is based upon the assumption that these structures are melanosomes. Though this assumption is consistent with the current data, chemical studies (such as refs.[25,54,55]) could provide further tests

of the identity of the structures. If future data support an alternative interpretation of the nanostructures (eg, as fossilized bacteria), then our inferences concerning colour will be invalidated.

## Methods

**Material and its provenance and access**. PMoL-B00175 preserves a nearly complete skeleton with associated feathers (Fig. 1). It was collected by a local farmer from Qinglong County, Hebei Province, China, and acquired by the Paleontological Museum of Liaoning in February, 2014. According to the collector, the specimen was from a quarry near Nanshimenzi Village in Gangou Township, Qinglong County, Hebei Province where the Upper Jurassic Tiaojishan Formation are exposed[56] and this was confirmed by our field investigation in 2016 (Supplementary Note 1). The specimen is now housed at the Paleontological Museum of Liaoning in accordance with local regulations, and it is available for public viewing and scientific study.

**Fossil preparation**. PMoL-B00175 was prepared using standard methods. Specifically, the preparation was carried out using pneumatic tools in a preparation chamber fitted with a stereomicroscope.

**Morphological study**. Standard morphological methods were used to study the new specimen, including observing the fossil by naked eyes and optic microscope, measuring the specimen using caliper, and comparing the specimen with other fossils through first-hand studies and literature-based study.

**Phylogenetic analysis**. The phylogenetic analysis was performed first using a data set modified from a recently published study[3] with several taxa including *Caihong* added in. The data matrix was edited in Winclada (ver 1.00.08) and analysed using TNT software package[57]. A second phylogenetic analysis was performed using an additional data set with increased taxonomic sampling[58]. The results of the two analyses are in general the same, though some minor differences exist (Supplementary Note 3), but which have no impact on the conclusions drawn in this study.

**Melanosome and fossil nanostructure measurements**. We obtained data on nanostructure or melanosome morphology from the fossil and extant bird feathers (see Supplementary Table 2 for taxa added to the data set of ref.[59]) using techniques previously described in ref.[34]. Briefly, samples from the fossil and extant bird species known to have iridescent colouration or platelet-shaped melanosomes (Supplementary Table 2) were imaged with a scanning electron microscope (Ziess Supra 40VP SEM) and analysed in ImageJ to obtain measurements of length (long axis of fossil nanostructures or melanosomes in the plane of the image) and

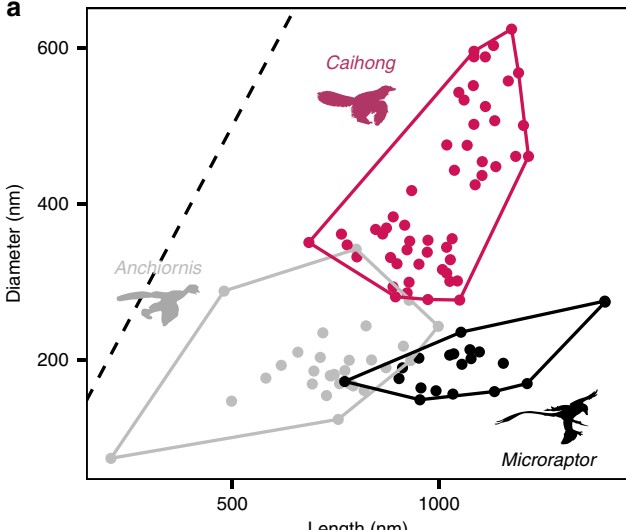

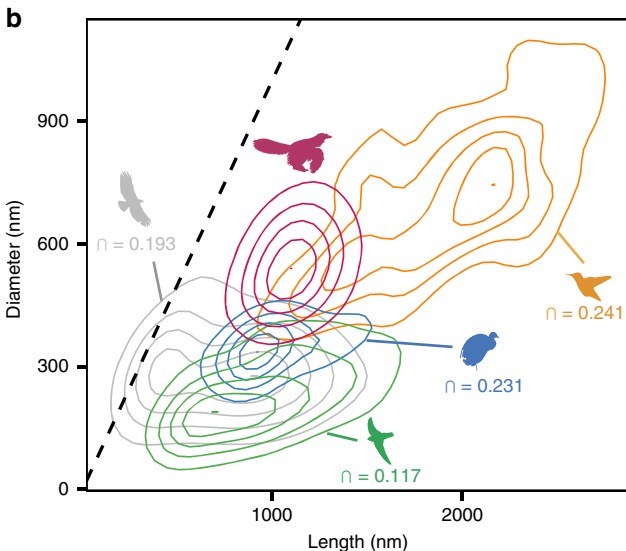

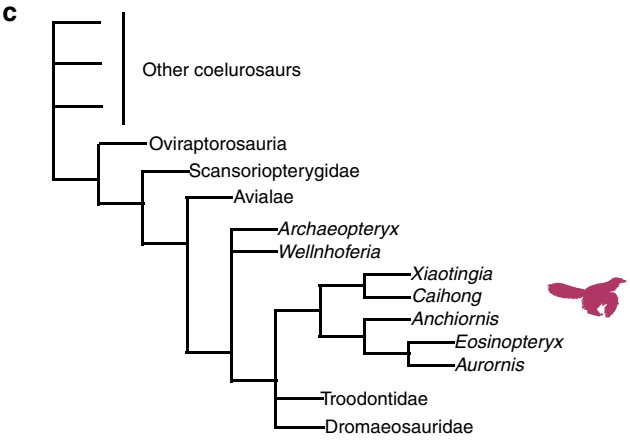

**Fig. 6** Analyses of nanostructure shape diversity and phylogenetic position of *Caihong juji*. **a** Mean length and diameter measurements for nanostructures from various body regions in *Caihong juji* (red), *Anchiornis huxleyi* (grey) and *Microraptor* (black). Rarefaction analyses show that sampling of the fossil taxa is appropriate to determine significant differences: *Anchiornis* shows significantly more nanostructure diversity than either *Caihong* or *Microraptor* (Supplementary Fig. 13). **b** Morphospace plots of raw nanostructure measurements for *Caihong juji* relative to extant avian groups with platelet-like melanosomes (orange: hummingbirds, green: swifts, blue: trumpeters and trogons, grey: other non-platelet avian melanosomes). Contour lines show 2D density of measurements. Numbers give proportional overlap (∩) between *Caihong juji* (red) and each group (Methods). **c** Simplified coelurosaurian phylogeny (strict consensus of 192 most parsimonious trees from the primary analysis) showing the recovered position of *Caihong juji* (Supplementary Note 3; Supplementary Figs. 18 and 19). Additional assessments of the taxon utilizing another recent data set recover an Anchiorninae clade as a part of Troodontidae (Supplementary Note 3; Supplementary Figs. 20, 22). Photo credits: Fred Wierum (*Microraptor*, CC BY SA 4.0, https://creativecommons.org/licenses/by-sa/4.0/legalcode), Dick Daniels (Trumpeter, CC BY SA 3.0, https://creativecommons.org/licenses/by-sa/3.0/legalcode), Alan Vernon (Red-tailed hawk, CC BY 2.0, https://creativecommons.org/licenses/by/2.0/legalcode), T R Shankar Raman (Glossy swiftlet, CC BY SA 4.0, https://creativecommons.org/licenses/by-sa/4.0/legalcode), Nobu Tamura (*Anchiornis*, CC BY 3.0, https://creativecommons.org/licenses/by/3.0/legalcode), J. Clarke (*Caihong*, modified from original artwork by Velizar Simeonovski, used with permission), Robert McMorran/USFWS (Anna's hummingbird, CC BY 2.0, https://creativecommons.org/licenses/by/2.0/legalcode)

diameter (short axis of fossil nanostructures or melanosomes measured along the image plane). Although melanosomes are clearly three-dimensional, and some nanostructures were observed at oblique angles (Fig. 5c), cross-sectional shape (perpendicular to the image) is difficult to observe with certainty from SEM images. Therefore, we also used FIB milling coupled with a tilting specimen holder (FEI) to cut cross-sections through individual nanostructures (Supplementary Figs. 8, 9) and/or image them from the side. In total, nanostructures were observed in 60 out of 66 fossil samples (images available on Figshare at 10.6084/m9.figshare.5427214 and 10.6084/m9.figshare.5427244). Measurements taken near internal organs/body cavity were excluded from the data set prior to analyses, resulting in a final data set of 53 locations in the fossil slab.

**Assessment of the impacts of taphonomic shrinkage on results**. High temperatures and pressures associated with fossil preservation might alter the morphology of melanosomes[47], but see ref. [60]. To test the possible effects of taphonomic shrinkage on our results, we re-ran key analyses assuming both length and diameter of fossil nanostructures had decreased by 20%[47] (see Supplementary Tables 3, 4; Supplementary Figs. 16, 17).

**Morphospace analyses**. Length and width for fossil nanostructures were compared to melanosome measurements for extant birds (taxonomic sample of ref. [59] plus new taxa listed in Supplementary Table 2). To compare nanostructure diversity in *Caihong* and other extinct paravian dinosaurs to that in extant birds, we used linear models to test whether the multivariate distance from the group mean centroid varied significantly between groups. Rarefaction of variance analyses were used to assess whether our sampling adequately captures variation (Supplementary Figs. 12, 13). To calculate the overlap of distributions in two-dimensional (2D) morphospace between the fossil measurements and other avian groups, we calculated 2D kernel densities with the kde package in R. To make 2D density estimates comparable among groups, we used a similar grid of points and bandwidth estimated from our data set, including the fossil. Two-dimensional densities were converted to proportions by dividing by the sum of each data set. Proportional overlap between the fossil and different groups was determined as the sum of the minima at each grid point (ie, if the proportional density for hummingbirds is zero for a region where the fossil density is 0.5, the probability would that the two distributions overlap would be zero at that specific length-diameter coordinate).

**Identifying platelet-shaped nanostructures**. We assessed whether (i) measurements for each sample fell outside of an extant avian morphospace exclusive of platelet-shaped melanosomes and (ii) SEM images showed evidence for overlapping mouldic nanostructures (Fig. 5a; Supplementary Fig. 7)—a pattern expected from thin, stacked platelets but not elliptical or spherical melanosome morphologies—or flattened, platelet-like 3D solid nanostructures (Fig. 5b–d; Supplementary Fig. 7). If both were true, the sample was classified as having platelet-shaped nanostructures. If both were false, the sample was classified as having typical spherical or rod-like melanosomes found in most extant birds. If the classification was ambiguous or mixed (ie, showing evidence for both spherical or elliptical nanostructures and platelet-shaped nanostructures), we ran a clustering analysis to classify only nanostructures significantly outside the densest region of avian morphospace as platelets and those overlapping avian morphospace as non-platelet elliptical or spherical. Nanostructures inferred as non-platelet-shaped were used in subsequent quadratic discriminant analyses (QDAs) to predict feather colour.

**Discriminant function analyses and plumage colour reconstruction**. To classify the colours of fossil samples, we ran a QDA following[34,35] in R ver. 3.2.2[61]. Briefly, we built a full model including all potential predictor variables (length, length CV, diameter, diameter CV, aspect ratio, aspect ratio skew, length skew and diameter skew); we used a backward stepwise approach to eliminate variables that did not significantly predict variation in colour ($p < 0.05$); and we used a QDA (accounting for multicollinearity among variables) to infer the colour of fossil feather samples. Previous analyses revealed that melanosome density is not linked to discrete colour classes[34,35], therefore we did not include density as a variable in these analyses. We ran analyses using six separate training data sets: (1) a data set with black, brown, grey and iridescent categories[34,35,59] in addition to a 'platelet iridescence' category (including all new taxa with platelet-shaped melanosomes); (2) a data set similar to data set 1 but with hummingbird platelets as a distinct group; (3) a data set similar to 1 and 2 but with non-hummingbird platelets grouped with other species with rod iridescence; (4) a data set similar to 1 but only classifying *Caihong* samples not showing evidence for platelet morphology and dense packing (eg, Supplementary Fig. 7); (5) a data set where brown-black colours in penguins, known to be linked to novel, large and spherical melanosomes not seen in other extant Aves[45] were assigned to a distinct group and only non-platelet *Caihong* samples were categorized; and (6) a data set that accounted for 20% taphonomic shrinkage[47] of fossil melanosomes (with the same categories as data set 1). The variables retained for use in the analysis were melanosome or nanostructure aspect ratio, length, diameter, length coefficient of variation CV, diameter CV and aspect ratio skew (all $p < 0.001$). The discriminant functions were highly significant for both training data sets (Supplementary Table 3). To assess the effectiveness of the discriminant

function at categorizing unknown colours based on morphology, we determined the proportion of feather samples with known colours that were categorized correctly (self-test) and also the performance of the model classifying colour when one sample was left out of the analysis (cross-validation; see Supplementary Table 3).

**Nomenclatural acts**. This published work and the nomenclatural acts it contains have been registered in ZooBank, the proposed online registration system for the International Code of Zoological Nomenclature. The ZooBank life science identifiers can be resolved and the associated information viewed by appending the life science identifiers (LSID urn:lsid:zoobank.org:pub: E2751CAA-BDCA-4367-B7E2-15A9812C6190) to the prefix http://zoobank.org/.

**Data availability**. The data reported in this paper are detailed in the main text and its supplementary information files. SEM images of *Caihong juji* and the extant birds listed in Supplementary Table 2 are available at 10.6084/m9.figshare.5427214 and 10.6084/m9.figshare.5427244.

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

## Acknowledgements

We thank Sun Ge for help during the course of the work, Zhang Lijun and Wang Yuan for discussion; Xiaoqing Ding and Matthew Brown for preparing the specimen; and Liesbet Van Landschoot for help with FIB/SEM. This work was supported by grants from NSFC (41172026, 41688103, 41120124002, 91514302 and 41272031), NSF EAR 1251922 and EAR 1251895, AFOSR FA9550-16-1-0331, HFSP RGY83 and LR2012038.

## Author contributions

D.H., X.X. and J.A.C. designed the project, D.H., X.X., J.A.C., C.M.E., R.Q., Q.L., M.D.S., C.Z., L.D.'A., and J.J. performed the research, and X.X., J.A.C., D.H., C.M.E., and M.D.S. wrote the manuscript.

## Additional information

**Competing interests:** The authors declare no competing financial interests.

