## [Peer Review File · Nature Communications]

Reviewers' comments:

Reviewer #1 (Remarks to the Author):

The authors describe a new taxon of very bird-like dinosaur from northeastern China, which they call *Caihong juji*. Many feathered dinosaurs have come out of China recently, and show no signs of abating. But this one stands out and is clearly an important find. It provides the earliest evidence of flattened melanosomes in dinosaurs (associated with iridescence and variable/vivid hues in modern birds), the earliest possible evidence for an alula, and has a noteworthy combination of a fairly short arm, a forearm proportionally longer than the humerus, and a huge tail with long asymmetrical feathers. This is something a novel bauplan among early birds and their closest dinosaur relatives. It demonstrates that the new fossil is indeed a new taxon, and also provides more evidence of incredible diversity in flight and feather structures around the origin of flight. Strangely, the new animal also has a fairly large 'horn' on its lacrimal, a cranial display structure previously unknown in very bird-like dinosaurs. It is further evidence that the fossil is a distinct taxon, and also that these dinosaurs on the cusp of becoming birds were using both their bones and their feathers for display. All-in-all, this is an important find that further clarifies (or, in fact, muddies a little—but that is ok!) our understanding of the origin of birds and flight. It deserves a high profile publication.

I am pleased to see this paper resubmitted. I reviewed it for another journal and gave it a very good review and recommended publication, and was surprised that it was not published there. The authors took into account all comments in my original review, so I little else to recommend. This paper can essentially be published as-is. However, I will make one final suggestion: Nature Communications allows slightly longer papers than Nature and Science, but the manuscript is still written to the very short length of these journals. I suggest that the authors move some of the morphological description from the supplement into the main paper. That will ensure that readers see it. Also, the skull line drawing would work well in the main text.

Steve Brusatte

Reviewer #2 (Remarks to the Author):

By tradition, the paleontological community is more forgiving towards 'liberal' interpretations than are scientists of other disciplines, particularly within natural sciences. However, we still need to base our conclusions on some sort of data. Hu et al. make a lot of claims pertaining to ancient colors and their role in sexual signaling; however, these are largely unsupported because they are based on the following (untested) assumptions:

1. The microbodies (and imprints thereof) found within the holotype of *Caihong* are all remnant melanosomes.
2. If indeed fossilized pigment organelles, then there are no contributions from internal

melanosomes.

3. The coloration of Caihong was produced solely by melanosomes and melanic pigments.

4. The flattened microbodies all represent platelet-shaped melanosomes.

Regarding the melanosome identification issue: Most (or all) microbodies described by Hu et al. could potentially be remnant pigment organelles, but no evidence is presented in support of this conclusion. Proper identification of fossil microbodies is a vehemently debated subject (see e.g. Moyer et al. 2014; Schweitzer et al. 2015; Vinther 2016). Likewise, the interpretation of microbodies and their imprints in feathered dinosaurs is not a straightforward process (Lindgren et al. 2015a). Nonetheless, means exist to distinguish remnant melanosomes from other types of microbodies, such as e.g. microorganismal cells (e.g. Glass et al. 2012; Lindgren et al. 2015b). Although pretty liberal interpretations of both IR microspectroscopic and ToF-SIMS data recently have been published, careful examination using these methods and/or (for instance) alkaline hydrogen peroxide oxidation could help identifying the fossil microbodies. Why have not any of these (or other) methods been applied to the fossil material?

It is assumed that all microbodies are epidermal rather than internal; however, internal melanosomes are abundant in modern animals. Moreover, they can fossilize (McNamara et al. 2014, 2016) and thus may be present within the body outline/cavity of Caihong. Means exist to chemically characterize the matrix in which the microbodies are embedded (e.g. Pan et al. 2016). Why has not this been done?

The reconstructed coloration of Caihong is based on the assumption that other than melanosomes, no other structures and/or pigments were originally present in the plumage of PMoL-B00175; however, in modern birds the coloration is the result of a variety of factors, including diet, keratin structure and co-expressed biochromes (e.g. Jawor & Breitwisch 2003). Why are these uncertainties not mentioned?

Regarding the 'platelet-shaped melanosomes': The entire fossil is flattened as a result of compaction, and thus it is very likely that the same diagenetic process has affected also those microstructures preserved within it. Normally, preservation is highly variable, even at the microscale. For instance, this can be seen in a specimen of *Anchiornis* where differently preserved microstructures occur immediately adjacent to one another (see Lindgren et al. 2015a, fig. 2a). Many of these have a flattened appearance, very similar to what is seen in Caihong. Hu et al. state that 'Compression has not been supported empirically or theoretically as taphonomic outcome', with a reference to McNamara et al. (2013); however, compression/compaction from one direction (i.e. what happens during burial) was not tested in that study.

Finally, it is stated that 'the fossilized melanosomes show no evidence of hollowness'. The possible presence/absence of internal structures and/or cavities cannot be assessed using a focused ion beam because it tends to melt the organic remains (as can be seen in Figure S12). My suggestion is that the authors instead try transmission electron microscopy (TEM),

which retains the structural integrity of the microbodies (when embedded in epoxy resin).

References

Glass, K. et al. 2012: Direct chemical evidence for eumelanin pigment from the Jurassic period. *Proc. Natl. Acad. Sci.* 109, 10218–10223.

Jawor J.M. & Breitwisch R. 2003: Melanin ornaments, honesty, and sexual selection. *Auk* 120, 249–265.

Lindgren, J. et al. 2015a: Molecular composition and ultrastructure of Jurassic paravian feathers. *Sci. Rep.* 5, 13520.

Lindgren, J. et al. 2015b: Interpreting melanin-based coloration through deep time: a critical review. *Proc. R. Soc. B* 282, 20150614.

McNamara M.E. et al. 2013 Experimental maturation of feathers: implications for reconstructions of fossil feather colour. *Biol. Lett.* 9, 20130184.

McNamara, M.E. et al. 2014: Non-integumentary melanosomes can bias reconstructions of the colours of fossil vertebrate skin. Paper presented at 4th International Palaeontological Congress. The history of life: a view from the Southern Hemisphere, Mendoza, Argentina. Abstract volume: Ianigla, CCT-Conicet (September 28–October 3).

McNamara, M.E. et al. 2016: Fossilization of melanosomes via sulfurization. *Palaeontology* 59, 337–350.

Moyer, A.E. et al. 2014: Melanosomes or microbes: testing an alternative hypothesis for the origin of microbodies in fossil feathers. *Sci. Rep.* 4, 4233.

Pan, Y. et al. 2016: Molecular evidence of keratin and melanosomes in feathers of the Early Cretaceous bird *Eoconfuciusornis*. *Proc. Natl. Acad. Sci.* 113, E7900–E7907.

Schweitzer, M.H. et al. 2015: Melanosomes and ancient coloration re-examined: a response to Vinther 2015 (doi 10.1002/bies.201500018). *BioEssays* 37, 1174–1183.

Vinther, J. 2016: Fossil melanosomes or bacteria? A wealth of findings favours melanosomes: Melanin fossilises relatively readily, bacteria rarely, hence the need for clarification in the debate over the identity of microbodies in fossil animal specimens. *BioEssays* 38, 220–225.

Reviewer #3 (Remarks to the Author):

The authors describe a new, well-preserved specimen of bird-like theropod dinosaur that preserves a significant part of the feather body covering. They also infer part of the plumage color using a methodology previously used (by some of the authors and others) for

other feathered dinosaurs. The authors document for the first time in the fossil record the presence of platelet-shaped melanosomes.

This study is particularly significant among theropod and bird palaeontologists as it adds a new taxon close to the root of bird origins. In particular, this new taxon differs from all other Jurassic paravians in its combination of features (but see comment below on *Pedopenna*). The dentition (and serration pattern) is unique among Jurassic paravians, which has interesting implications on the discussion on the palaeoecological context of bird origins. This was briefly mentioned by the authors, who have focused on plumage and ornamentation (see my comment below): the palaeoecological implications of this new dental pattern among Jurassic paravians should be further discussed as they may result particularly interesting among the broader palaeontological and biological community.

I have three main questions on this manuscript that would appreciate that the authors evaluate and discuss:

1-Anatomical description:

My main concern relates to the claimed presence of a bony crest in this taxon. This is particularly relevant because the authors name the new species according to this feature, include it in the diagnosis of the taxon (and even mention it in the title of the manuscript). The authors report that the specimen bears the "lacrima with [a] prominent dorsolaterally oriented crest", a feature previously unreported among Jurassic paravians, and usually absent among maniraptoran theropods. Lacrimal crests are widely distributed among basal (non-maniraptoran) theropods, but in paravians (as the authors note) lacrimal ornamentation is limited to lateral shelves along the anterior ramus, overhanging the antorbital fossa. A dorsolaterally oriented crest is thus an unusual (and unexpected) feature among these close relatives of birds.

Although I have no a priori objections against a lacrimal crest in a basal paravian, I suspect this is a preservational artifact, and would like the authors to more carefully discuss this and add further evidence that it is a genuine biological feature. This is necessary given that, as I noted above, this feature is so relevant in the diagnosis (and in the species name) of the taxon.

Before listing my reasons for asking a careful discussion of the lacrimal crest, I remark that even if this crest turns out to be a preservational artifact, the validity of this new species is not affected: the specimen shows several unique features in the premaxilla, maxillary pneumatization, dentition, limb proportions that strongly support its status as a new valid species. Thus, the presence (or absence) of a bony crest is, in my opinion, not significant for the value and significance of this manuscript. But this absence/presence must be clearly and unambiguously supported in the manuscript in its actual form.

My suspicion that the bony projection in the lacrima is an artifact is based on these lines of evidence (see also Fig. S5):

- the orbital and postorbital parts of the skull are clearly dorsoventrally compressed: the palatine is dorsally displaced inside the antorbital fenestra, the postorbital broadly overlaps the jugal, the tip of the ascending ramus of jugal reaches the dorsal margin of the infratemporal fenestra. The skull appears quite long and low, differing from the more triangular outline in other basal paravians: given the displacement of the above mentioned

bones, this shape is clearly preservational. This means that we cannot exclude that some unusual elements like the lacrimal crest are preservational artifacts due to deformation and displacement of bones.

- the element that the authors consider as the posterior ramus of the lacrimal might be the anterior end of the frontal. This element forms an acute corner with the ventral lacrimal ramus: this is unusual when compared to other basal paravians, where the posterodorsal ramus of lacrimal forms an obtuse corner with the rest of the lacrimal, and usually forms a broad arch of the large orbital fenestra. Such narrow anterodorsal corner of the orbit is clearly a taphonomic artifact, which indicates that the relative positions of the elements in that part of the skull do not show the original shape. Unfortunately, the Fig. S5 is not clear, but I note that the element claimed to be the posterodorsal ramus of lacrimal is aligned to the displaced frontal bone, with a large crack crossing between them: I suspect that the two elements are just parts of the same bone (the frontal): this is consistent with the larger contribution to the orbit made by the frontal among all other paravians.

In conclusion, I suspect that the "lacrimal crest" is simply the dorsal end of the preorbital ramus of the lacrimal as in all other paravians, and that no dorsolateral crest was present in this taxon. Accordingly, I suggest to not name the species according to this problematic element, and to remove this feature from the diagnosis of the species (and from the title of the manuscript).

I may be wrong, I have based my arguments exclusively on the images and information provided by the authors: if they consider the lacrimal crest as a genuine feature, I would appreciate much additional evidence for supporting this. Therefore, my main request to the authors is to provide additional anatomical description and close-up images of the lacrimal (on both slabs and with different light angles) because this element is a pivotal feature in their diagnosis of the new species (and is even mentioned in both species name and manuscript title!).

As I noted above, even in absence of a bony crest this is a significant and interesting specimen: its removal from the diagnosis and description is in my opinion a relatively minor change to the manuscript.

2-Reproducibility of the results:

I replicated their main phylogenetic analysis (based on Xu et al. 2015), using the data matrix provided, and confirm the topology and tree statistics they discuss. Unfortunately, I was not able to replicate the second analysis (based on Brusatte et al. 2014). The authors stated that the analysis produced 100000 shortest trees of 3514 steps, where Caihong is found as sister taxon of Xiaotingia among Anchiornithinae. My re-analysis, using the scores provided by the authors for Caihong and using the data set of Brusatte et al. (2014) resulted in 100000 shortest trees of 3399 steps long (much shorter than the value obtained by the authors): in the strict consensus tree, Caihong results among Anchiornithinae but in a not-resolved polytomy. I have tested different settings (in particular, setting all characters as non-additive, as in Xu et al. 2015), but have not being able to recover the results reported by the authors.

Please, provide a complete matrix for the second analysis (and the character setting) to allow replication of their topology.

A note on the two phylogenetic analyses used. Xu et al. (2015) and Brusatte et al. (2014) used comparable sets of character statements, but differ each other in the use of additive (ordered) characters (not used by Xu et al., explicitly used by Brusatte et al.). This means that the two analyses follow different interpretations of the state transitions for the same characters.

I encourage the authors to follow only one of the two interpretations, as they contradict each other in the homology assumptions.

In my opinion, the option followed by the analysis in Brusatte et al. (2014) is more realistic, as it assumes nested homologies among alternative states of some multistate characters. I re-tested the matrix based on Xu et al. (2015) re-setting the state transitions among those multistate characters clearly defined as additive (the approach followed in Brusatte et al. 2014), and included the following string (for TNT file) to those characters that form nested sets of transitions (and thus could be considered as additive):

```
ccode + 15 16 17 37 40 45 64 65 68 76 83 109 112 115 118 120 122 141 153 156 163 166  
171 174 175 199 216 228 273 276 297 298 301 311 319 321 330 334 344;
```

[Note that in TNT, character numeration starts from 0, thus - for example - "character 15" in the above string is "character 16" in the list of Xu et al. 2015]

The result of this alternative analysis obtained a slightly different topology than the one discussed by the authors: Anchiornithinae results the sister-taxon of Dromaeosauridae+Troodontidae. This example shows how a priori assumptions on character transition settings may produce different topologies of the same data matrix.

I encourage the authors to clearly state the hypotheses followed in their phylogenetic analyses, and to consistently follow the same setting on both analyses.

3-Comparison with Pedopenna:

Pedopenna is based on a fragmentary specimen preserving exclusively the tibiotarsus and pes. The authors stated that Caihong differs from Pedopenna in the relatively shorter first metatarsal (<15% of metatarsus length, compared to 25% in Pedopenna) and in "extensively feathered pedal phalanges". According to Xu and Zhang (2005: A new maniraptoran dinosaur from China with long feathers on the metatarsus.

Naturwissenschaften 10.1007/s00114-004-0604-y), the first metatarsal of Pedopenna is not complete, but the preserved element is <10% of metatarsal III: this ratio is comparable to the condition reported in Caihong, and differ from other basal paravians. If the authors have new information on Pedopenna, this should be explicitly stated, otherwise, this feature actually cannot differentiate Caihong from Pedopenna.

The second difference may be preservational, as documented among the different specimens of Anchiornis.

So, apparently there are no actual features differentiating Caihong from Pedopenna.

Nevertheless, Pedopenna is diagnosed by a very slender pedal phalanx I-1 (length/mid-shaft diameter ratio about 7.2): please, provide information on this element in Caihong since it may indicate whether it is distinct from Pedopenna.

Give the possible synonymy with Pedopenna, I encourage the authors to include Pedopenna in the data set of Xu et al. (2015) to test this hypothesis. Although Pedopenna is already included in the analysis of Brusatte et al. (2014, where it results a scansoriopterygid), the latter data set is not designed to test basal paravian relationships (it mostly focuses on basal coelurosaurs): in overall features Pedopenna is clearly an Anchiornis-grade paravian, as is implicitly assumed also by the authors of this study who included Pedopenna in the differential diagnosis of Caihong.

In conclusion, this is a well-written manuscript describing a new and interesting taxon of paravian theropod.

The amount of modification to this manuscript from its actual form before final acceptance is relatively minor, and I see no reasons for not accepting it for publication once the above listed points have been discussed and implemented.

Andrea Cau
Geological Museum "G. Capellini", Bologna (Italy)

We have revised the ms based on the referees' helpful comments. Most notably, we:

1. Moved the description section into the main article
2. Added more description of the lacrimal and particularly of the lacrimal crest
3. Revised the diagnosis
4. Added new phylogenetic analyses and revised relevant discussions
5. Added some discussions to, and made our assumptions clearer in, the melanosome section.

We thank the referees for their constructive comments that have greatly improved the ms. We have toned down all conclusions from the microstructures. We have done a truly massive study with novel statistical approaches and assessment of morphospaces of fossils in the same units. We have added extensive new data on extant birds as well as now run 3 different phylogenetic analyses at the request of other referees.

While we absolutely appreciate the immense value of a separate chemical study in the referee's (2) desired framework, and would be happy to collaborate with them to do this in the future, we stand with the value of the extensive work that we have done in its own right—although with major changes to wording to address concerns with identification of the structures.

Responses to the referees:

Reviewer #1 (Remarks to the Author):

*The authors describe a new taxon of very bird-like dinosaur from northeastern China, which they call *Caihong juji*. Many feathered dinosaurs have come out of China recently, and show no signs of abating. But this one stands out and is clearly an important find. It provides the earliest evidence of flattened melanosomes in dinosaurs (associated with iridescence and variable/vivid hues in modern birds), the earliest possible evidence for an alula, and has a noteworthy combination of a fairly short arm, a forearm proportionally longer than the humerus, and a huge tail with long asymmetrical feathers. This is something a novel bauplan among early birds and their closest dinosaur relatives. It demonstrates that the new fossil is indeed a new taxon, and also provides more evidence of incredible diversity in flight and feather structures around the origin of flight. Strangely, the new animal also has a fairly large 'horn' on its lacrimal, a cranial display structure previously unknown in very bird-like dinosaurs. It is further evidence that the fossil is a distinct taxon, and also that these dinosaurs on the cusp of becoming birds were using both their bones and their feathers for display. All-in-all, this is an important find that further clarifies (or, in fact, muddies a little—but that is ok!) our understanding of the origin of birds and flight. It deserves a high profile publication.*

I am pleased to see this paper resubmitted. I reviewed it for another journal and gave it a very good review and recommended publication, and was surprised that it was not published there. The authors took into account all comments in my original review, so I little else to recommend. This paper can essentially be published as-is. However, I will make one final suggestion: Nature Communications allows slightly longer papers than Nature and Science, but the manuscript is still written to the very short length of these journals. I suggest that the authors move some of the morphological description from the supplement into the main paper. That will ensure that readers see it. Also, the skull line drawing would work well in the main text.

We thank the referee for his positive comments. Following his suggestion, we have moved the morphological description and several illustrations including the skull line drawing into the main article.

Reviewer #2 (Remarks to the Author):

By tradition, the paleontological community is more forgiving towards ‘liberal’ interpretations than are scientists of other disciplines, particularly within natural sciences. However, we still need to base our conclusions on some sort of data.

We understand the referee’s concerns with some practices of the paleontological community. However, we have collected large amounts of data for this paper from both the fossil and from extant birds, including hundreds of new SEM images (fossil and new extant taxa) and thousands of measurements. We have then analyzed these data using rigorous statistical analyses that are not typically seen in the paleontological literature. So to say that our interpretations need “some sort of data” to back them up does not seem to be an entirely accurate appreciation of our work.

Hu et al. make a lot of claims pertaining to ancient colors and their role in sexual signaling; however, these are largely unsupported because they are based on the following (untested) assumptions:

One cannot do palaeontology (or science in general) without making assumptions. However, as long as these assumptions are reasonable, backed by data or logic and are clearly stated, this is acceptable and should not prevent rigorous work from moving forward. In particular, when an assumption has been supported by tests from numerous independent groups (including by those who have been most skeptical of it), it is reasonable to partially rely on these results and not repeat them for every new case. Science works by building on previous findings, not endlessly repeating them. Otherwise, we would have a lot expense for very little benefit. In this case, we argue that our assumptions are backed by data and logic (see details below), but agree that in some cases we did not clearly state them.

We tone down our claims throughout by making our assumptions clear, addressing the referees concerns and cutting some text.

- 1. The microbodies (and imprints thereof) found within the holotype of Caihong are all remnant melanosomes. Regarding the melanosome identification issue: Most (or all) microbodies described by Hu et al. could potentially be remnant pigment organelles, but no evidence is presented in support of this conclusion. Proper identification of fossil microbodies is a vehemently debated subject (see e.g. Moyer et al. 2014; Schweitzer et al. 2015; Vinther 2016). Likewise, the interpretation of microbodies and their imprints in feathered dinosaurs is not a straightforward process (Lindgren et al. 2015a). Nonetheless, means exist to distinguish remnant melanosomes from other types of microbodies, such as e.g. microorganismal cells (e.g. Glass et al. 2012; Lindgren et al. 2015b). Although pretty liberal interpretations of both IR microspectroscopic and ToF-SIMS data recently have been published, careful examination using these methods and/or (for instance) alkaline hydrogen peroxide oxidation could help identifying the fossil microbodies. Why have not any of these (or other) methods been applied to the fossil material?*

We agree that our wording was perhaps too strong and we have now modified the main text to make clear these are microstructures, identified by us as melanosomes, that that is our assumption.

Although this is certainly an assumption, it is based on data, the morphological similarity of these microstructures to modern melanosomes (see figure 3, SOM) and on the large amount of morphological and chemical data in the literature that universally support melanosome identity (e.g. Pan et al. 2017, Lindgren et al. 2014, Colleary et al. 2015, Peteya et al. 2016). It is important to note that this work using chemical data to assess melanosome identity has been done by five independent groups (Roger Simon's group at Duke, Mary Schweitzer and Johan Lindgren's groups at UNC and Lund, Roy Wogelius and Phil Manning's group at Manchester, Jakob Vinther's group at U. Bristol and our own group) and that one of them (Schweitzer in particular and Lindgren to a lesser extent) has been skeptical of the melanosome hypothesis. Thus, these previous chemical results supporting melanosome identity cannot be attributed to bias of a certain research group. Moreover, despite extensive commentary (e.g. Moyer et al. 2014, Barden et al. 2015, Lindgren et al. 2015, Schweitzer et al. 2016), no paper has provided any positive, quantitative evidence that microbodies are bacteria or anything other than melanosomes. By contrast, numerous studies have shown quantitative data of using different techniques (ToF-SIMS, Raman, pump-probe spectroscopy, FTIR, etc.) in support of the melanosome hypothesis. We thus consider it safe to assume at this point that they are indeed fossilized melanosomes, but are open to reconsideration if any evidence of e.g. bacterial identity is presented in the literature.

In summary, we appreciate the chemical work suggested by the referee but argue that, given 1) the morphological similarity of these microbodies to melanosomes, 2) previous morphological and chemical evidence in support of this interpretation and 3) the lack of support for any alternative interpretation, they are not needed in this case.

However, throughout the paper, we now make it clear that this is an assumption, refer to them as "structures" and generally tone down our wording.

2. *If indeed fossilized pigment organelles, then there are no contributions from internal melanosomes.*

It is assumed that all microbodies are epidermal rather than internal; however, internal melanosomes are abundant in modern animals. Moreover, they can fossilize (McNamara et al. 2014, 2016) and thus may be present within the body outline/cavity of Caihong. Means exist to chemically characterize the matrix in which the microbodies are embedded (e.g. Pan et al. 2016). Why has not this been done?

We argue that this is a safe assumption because we sample exclusively from the feathers of this fossil. It is extremely unlikely that melanosomes migrated from the internal organs through the body and into the feathers while demarcating the fine scale structure of the feather. In the unlikely event that internal melanosomes migrated outside of the body, we would expect them to randomly aggregate into a "halo" or pools (which are not seen in this fossil), and not infiltrate the feathers. Furthermore, although data on shape and size of melanosomes in internal organs are limited, we know of no evidence that any of them are flattened as are the ones we report here. We now note this in the main text.

3. *The coloration of Caihong was produced solely by melanosomes and melanic pigments.*

The reconstructed coloration of Caihong is based on the assumption that, other than melanosomes, no other structures and/or pigments were originally present in the plumage of PMoL-B00175; however, in

modern birds the coloration is the result of a variety of factors, including diet, keratin structure and co-expressed biochromes (e.g. Jawor & Breitwisch 2003). Why are these uncertainties not mentioned?

We appreciate this point, and indeed have addressed it in previous work (Li et al. 2012). However, we do not need to make this assumption here, as we do not attempt to assign specific colors to this fossil. Rather, our focus in this portion of the paper is on the presence of the unusual melanosome morphologies that are similar to those seen in some extant iridescent feathers. These indicate that the feathers likely had some form of iridescence, but the precise color is unknowable at this point, as we note.

Two of the authors (MDS and LD) have recently written a comprehensive review paper on mixed structural and pigmentary colors (Shawkey and D’Alba 2017), and found no example of an iridescent feather containing any additional pigments. Moreover, the presence of an additional pigment would not change the fact that these unusual morphologies are present in the fossil.

Furthermore, the color of the non-iridescent feathers predicted as black in this fossil are unlikely to be affected by the presence of pigments, as these would be negated by the strong light absorption of melanin. For example, Hofmann et al. (2007) showed that orange carotenoids were present in black feathers of several oriole species, but had no effect on color. We have noted this pattern in our previous work (Li et al. 2012), but repeat it here for clarity.

4. The flattened microbodies all represent platelet-shaped melanosomes.

*Regarding the ‘platelet-shaped melanosomes’: The entire fossil is flattened as a result of compaction, and thus it is very likely that the same diagenetic process has affected also those microstructures preserved within it. Normally, preservation is highly variable, even at the microscale. For instance, this can be seen in a specimen of *Anchiornis* where differently preserved microstructures occur immediately adjacent to one another (see Lindgren et al. 2015a, fig. 2a). Many of these have a flattened appearance, very similar to what is seen in *Caihong*. Hu et al. state that ‘Compression has not been supported empirically or theoretically as taphonomic outcome’, with a reference to McNamara et al. (2013); however, compression/compaction from one direction (i.e. what happens during burial) was not tested in that study.*

This is an interesting point that we previously addressed this point in the supplement. Other microstructures from the same specimen and from other specimens from the same locality with the same geologic history show no compaction or flattening. Thus we do not consider this likely. Even if true, such flattening would not alter the 2D structure of the melanosomes that is also consistent with the 2D structure of flattened melanosomes from extant birds.

We note that the apparently morphology of some melanosomes in *Anchiornis* mentioned by the reviewer do not show the organization or 2D morphology of the melanosomes in *Caihong*. We agree that McNamara does not test compaction in this paper and have therefore removed this statement, and have clarified our argument against the presence of internal melanosomes.

Finally, it is stated that ‘the fossilized melanosomes show no evidence of hollowness’. The possible presence/absence of internal structures and/or cavities cannot be assessed using a focused ion beam because it tends to melt the organic remains (as can be seen in Figure S12). My suggestion is that the authors instead try transmission electron microscopy (TEM), which retains the structural integrity of the microbodies (when embedded in epoxy resin).

We thank the referee for this suggestion. Since (1) we do not argue for hollowness, (2) hollowness, if present, would only make our results (affinities with hummingbird style iridescence) stronger and (3) hummingbird hollowness is clearly visible in SEM (see figure), these additional analyses would not affect our core results and are outside the purview of the work.

Reviewer #3 (Remarks to the Author):

The authors describe a new, well-preserved specimen of bird-like theropod dinosaur that preserves a significant part of the feather body covering. They also infer part of the plumage color using a methodology previously used (by some of the authors and others) for other feathered dinosaurs. The authors document for the first time in the fossil record the presence of platelet-shaped melanosomes.

This study is particularly significant among theropod and bird palaeontologists as it adds a new taxon close to the root of bird origins. In particular, this new taxon differs from all other Jurassic paravians in its combination of features (but see comment below on Pedopenna). The dentition (and serration pattern) is unique among Jurassic paravians, which has interesting implications on the discussion on the palaeoecological context of bird origins. This was briefly mentioned by the authors, who have focused on plumage and ornamentation (see my comment below): the palaeoecological implications of this new dental pattern among Jurassic paravians should be further discussed as they may result particular interesting among the broader palaeontological and biological community.

I have three main questions on this manuscript that would appreciate that the authors evaluate and discuss:

1-Anatomical description:

My main concern relates to the claimed presence of a bony crest in this taxon. This is particularly relevant because the authors name the new species according to this feature, include it in the diagnosis of the taxon (and even mention it in the title of the manuscript). The authors report that the specimen bears the "lacrima with [a] prominent dorsolaterally oriented crest", a feature previously unreported among Jurassic paravians, and usually absent among maniraptoran theropods. Lacrimal crests are widely distributed among basal (non-maniraptoran) theropods, but in paravians (as the authors note) lacrimal ornamentation is limited to lateral shelves along the anterior ramus, overhanging the antorbital fossa. A dorsolaterally oriented crest is thus an unusual (and unexpected) feature among these close relatives of birds.

Although I have no a priori objections against a lacrimal crest in a basal paravian, I suspect this is a preservational artifact, and would like the authors to more carefully discuss this and add further evidence that it is a genuine biological feature. This is necessary given that, as I noted above, this feature is so relevant in the diagnosis (and in the species name) of the taxon.

Before listing my reasons for asking a careful discussion of the lacrimal crest, I remark that even if this crest turns out to be a preservational artifact, the validity of this new species is not affected: the specimen shows several unique features in the premaxilla, maxillary pneumatization, dentition, limb proportions that strongly support its status as a new valid species. Thus, the presence (or absence) of a bony crest is, in my opinion, not significant for the value and significance of this manuscript. But this absence/presence must be clearly and unambiguously supported in the manuscript in its actual form.

My suspect that the bony projection in the lacrimal is an artifact is based on these lines of evidence (see also Fig. S5):

- the orbital and postorbital parts of the skull are clearly dorsoventrally compressed: the palatine is dorsally displaced inside the antorbital fenestra, the postorbital broadly overlaps the jugal, the tip of the ascending ramus of jugal reaches the dorsal margin of the infratemporal fenestra. The skull appears quite long and low, differing from the more triangular outline in other basal paravians: given the

displacement of the above mentioned bones, this shape is clearly preservational. This means that we cannot exclude that some unusual elements like the lacrimal crest are preservational artifacts due to deformation and displacement of bones.

- the element that the authors consider as the posterior ramus of the lacrimal might be the anterior end of the frontal. This element forms an acute corner with the ventral lacrimal ramus: this is unusual when compared to other basal paravians, where the posterodorsal ramus of lacrimal forms an obtuse corner with the rest of the lacrimal, and usually forms a broad arch of the large orbital fenestra. Such narrow anterodorsal corner of the orbit is clearly a taphonomic artifact, which indicates that the relative positions of the elements in that part of the skull do not show the original shape. Unfortunately, the Fig. S5 is not clear, but I note that the element claimed to be the posterodorsal ramus of lacrimal is aligned to the displaced frontal bone, with a large crack crossing between them: I suspect that the two elements are just parts of the same bone (the frontal): this is consistent with the larger contribution to the orbit made by the frontal among all other paravians.

In conclusion, I suspect that the "lacrimal crest" is simply the dorsal end of the preorbital ramus of the lacrimal as in all other paravians, and that no dorsolateral crest was present in this taxon. Accordingly, I suggest to not name the species according to this problematic element, and to remove this feature from the diagnosis of the species (and from the title of the manuscript).

I may be wrong, I have based my arguments exclusively on the images and information provided by the authors: if they consider the lacrimal crest as a genuine feature, I would appreciate much additional evidence for supporting this. Therefore, my main request to the authors is to provide additional anatomical description and close-up images of the lacrimal (on both slabs and with different light angles) because this element is a pivotal feature in their diagnosis of the new species (and is even mentioned in both species name and manuscript title!).

As I noted above, even in absence of a bony crest this is a significant and interesting specimen: its removal from the diagnosis and description is in my opinion a relatively minor change to the manuscript.

While we respect the referee's opinion, we do have strong evidence for the presence of prominent lacrimal crests. Indeed, it may difficult to understand the cranial morphology just based on illustrations, in particular given that the skull and mandible expose in an oblique way (ventrolaterally rather than laterally) and some cranial elements are displaced from their original anatomical positions and display many breakages. However, a close examination suggests that a prominent lacrimal crest is present anterodorsal to the left orbit (and presumably another one to the right orbit). The left lacrimal is tetra-radiated element (rather than tri-radiated as in most paravians except *Austroraptor*: besides the anterior, posterior, and descending processes, there is an additional process at the junction area of the three processes which extends laterally first and then curves dorsally. This process also has an expanded, horn-like shape, which is unlike any other lacrimal process. The posterior process has a gradual, continuous transition to the lacrimal crest, though the angle between the two processes is relatively sharp. Beneath the posterior process, there is a separate bone which represents the anterior portion of the frontal (sorry that in the previous line-drawing, we did not illustrate this, which we believe is the reason that the referee considers the possibility of the posterior process being a part of the frontal). We agree with the referee that the narrow anterodorsal corner of the orbit is a taphonomic artifact, resulting from multiple factors, including the deformations and slight displacement of the bone and visual effect by the ventrolateral exposure of the skull.

Nevertheless, in the revised version, we provide more detailed description of the lacrimal morphology, and particularly of the lacrimal crest morphology, and we also revised the line-drawing to more accurately reflect the shape of the lacrimal crest and its relationships to the

neighboring bones. We appreciate the referee's comments which make the description much clearer and we hope that the revised version will satisfy the referee.

2-Reproducibility of the results:

I replicated their main phylogenetic analysis (based on Xu et al. 2015), using the data matrix provided, and confirm the topology and tree statistics they discuss. Unfortunately, I was not able to replicate the second analysis (based on Brusatte et al. 2014). The authors stated that the analysis produced 100000 shortest trees of 3514 steps, where Caihong is found as sister taxon of Xiaotingia among Anchiornithinae. My re-analysis, using the scores provided by the authors for Caihong and using the data set of Brusatte et al. (2014) resulted in 100000 shortest trees of 3399 steps long (much shorter than the value obtained by the authors): in the strict consensus tree, Caihong results among Anchiornithinae but in a not-resolved polytomy. I have tested different settings (in particular, setting all characters as non-additive, as in Xu et al. 2015), but have not been able to recover the results reported by the authors. Please, provide a complete matrix for the second analysis (and the character setting) to allow replication of their topology.

A note on the two phylogenetic analyses used. Xu et al. (2015) and Brusatte et al. (2014) used comparable sets of character statements, but differ each other in the use of additive (ordered) characters (not used by Xu et al., explicitly used by Brusatte et al.). This means that the two analyses follow different interpretations of the state transitions for the same characters.

I encourage the authors to follow only one of the two interpretations, as they contradict each other in the homology assumptions.

In my opinion, the option followed by the analysis in Brusatte et al. (2014) is more realistic, as it assumes nested homologies among alternative states of some multistate characters. I re-tested the matrix based on Xu et al. (2015) re-setting the state transitions among those multistate characters clearly defined as additive (the approach followed in Brusatte et al. 2014), and included the following string (for TNT file) to those characters that form nested sets of transitions (and thus could be considered as additive):

*ccode + 15 16 17 37 40 45 64 65 68 76 83 109 112 115 118 120 122 141 153 156 163 166 171 174 175
199 216 228 273 276 297 298 301 311 319 321 330 334 344;*

[Note that in TNT, character numeration starts from 0, thus - for example - "character 15" in the above string is "character 16" in the list of Xu et al. 2015]

The result of this alternative analysis obtained a slightly different topology than the one discussed by the authors: Anchiornithinae results the sister-taxon of Dromaeosauridae+Troodontidae. This example shows how a priori assumptions on character transition settings may produce different topologies of the same data matrix.

I encourage the authors to clearly state the hypotheses followed in their phylogenetic analyses, and to consistently follow the same setting on both analyses.

We thank the referee for the comments, and following the referee's suggestion, we have provided a complete matrix for the second analysis. also following the referee's suggestion, we ran the third analysis on Brusatte et al 2014 matrix with some multistate characters ordered, and we have amended the text to discuss the result of unordering/ordering these characters.

As shown in the revised version of the submission, our analyses produced basically the same results as our previous version. However, with some characters ordered, the analysis did produce unresolved phylogenetic relationships among a monophyletic anchiornithinae. Although the referee prefers a strategy of using ordered characters, there is no real theoretical basis favoring ordered

characters rather than unordered characters. So called nest homologies can be arbitrarily determined as in practice even the same working group may choose different multistate characters to be ordered (e.g., see different studies by the AMNH group); furthermore, developmentally, some assumed nest homologies can be produced by separately without a transitional state.

Nevertheless, we have provided results from analyses with both ordered or unordered characters, and the readers can make a judgement what results are more realistic. Finally, the results from different analyses have no effect on the conclusions drew in our paper.

3-Comparison with Pedopenna:

Pedopenna is based on a fragmentary specimen preserving exclusively the tibiotarsus and pes. The authors stated that *Caihong* differs from *Pedopenna* in the relatively shorter first metatarsal (<15% of metatarsus length, compared to 25% in *Pedopenna*) and in "extensively feathered pedal phalanges". According to Xu and Zhang (2005: A new maniraptoran dinosaur from China with long feathers on the metatarsus. *Naturwissenschaften* 10.1007/s00114-004-0604-y), the first metatarsal of *Pedopenna* is not complete, but the preserved element is <10% of metatarsal III: this ratio is comparable to the condition reported in *Caihong*, and differ from other basal paravians. If the authors have new information on *Pedopenna*, this should be explicitly stated, otherwise, this feature actually cannot differentiate *Caihong* from *Pedopenna*.

The second difference may be preservational, as documented among the different specimens of *Anchiornis*.

So, apparently there are no actual features differentiating *Caihong* from *Pedopenna*. Nevertheless, *Pedopenna* is diagnosed by a very slender pedal phalanx I-1 (length/mid-shaft diameter ratio about 7.2): please, provide information on this element in *Caihong* since it may indicate whether it is distinct from *Pedopenna*.

Give the possible synonymy with *Pedopenna*, I encourage the authors to include *Pedopenna* in the data set of Xu et al. (2015) to test this hypothesis. Although *Pedopenna* is already included in the analysis of Brusatte et al. (2014, where it results a scansoriopterygid), the latter data set is not designed to test basal paravian relationships (it mostly focuses on basal coelurosaurs): in overall features *Pedopenna* is clearly an *Anchiornis*-grade paravian, as is implicitly assumed also by the authors of this study who included *Pedopenna* in the differential diagnosis of *Caihong*.

Following the referee's suggestion, we have provided additional evidence to distinguish the new taxon from *Pedopenna*. The differences (proportional and discrete) are clear.

Lindgren J, et al. Interpreting melanin-based coloration through deep time: a critical review. *P Roy Soc B-Biol Sci* **282**, 20150614 (2015).

Lindgren J, et al. Molecular composition and ultrastructure of Jurassic paravian feathers. *Scientific Reports* **5**, 13520 (2015).

Colleary C, et al. Chemical, experimental, and morphological evidence for diagenetically altered melanin in exceptionally preserved fossils. *Proceedings of the National Academy of Sciences of the United States of America* **112**, 12592-12597 (2015).

K. Glass et al., Impact of diagenesis and maturation on the survival of eumelanin in the fossil record. *Organic Geochemistry* **64**, 29-37 (2013).

Moyer, A.E., W. Zheng, E. A. Johnson, M. C. Lamanna, D. Li, K. J. Lacovara, and M. H. Schweitzer. Melanosomes or microbes: testing an alternative hypothesis for the origin of microbodies in fossil feathers. *Scientific Reports* 4. doi:10.1038/srep04233 (2014).

Glass, K., Ito, S., Wilby, P. R., Sota, T., Nakamura, A., Bowers, C. R., Vinther, J., Dutta, S., Summons, R., Briggs, D. E. G., Wakamatsu, K. and Simon, J. D. 2012. Direct chemical evidence for

- eumelanin pigment from the Jurassic period. *Proceedings of the National Academy of Sciences of the United States of America*, **109**, 10218-10223.
- Q. Li, K. Gao, Q. Meng, J.A. Clarke, **M.D. SHAWKEY***, L. D'Alba, R. Pei, M. Ellison, M.A. Norell, J. A. Vinther. A new specimen of *Microraptor* and the evolution of iridescent plumage color. *Science* 335:1215-1219 (2012).
- Hofmann, C. M., McGraw, K. J., Cronin, T. W. and Omland, K. E. Melanin coloration in New World orioles
I: carotenoid masking and pigment dichromatism in the Orchard Oriole complex. *Journal of Avian Biology* 38: 163-171 (2007).
- Moyer, A.E., W. Zheng, E. A. Johnson, M. C. Lamanna, D. Li, K. J. Lacovara, and M. H. Schweitzer. Melanosomes or microbes: testing an alternative hypothesis for the origin of microbodies in fossil feathers. *Scientific Reports* 4. doi:10.1038/srep04233 (2014).

Reviewers' comments:

Reviewer #3 (Remarks to the Author):

The authors have taken into account the suggestions for improving the descriptive part, and for allowing the replication of their results. Also, they have commented in detail to the questions raised during the first revision.

I suggest for the publication of the manuscript in its actual form.

Andrea Cau, PhD

Museo Geologico e Paleontologico "G. Capellini", Bologna, Italy

Reviewer #4 (Remarks to the Author):

See attached doc.

For Authors:

In this manuscript, the authors describe a new species of feathered theropod, proposed to be black and iridescent, from the late Jurassic of China with both macro (skeletal and plumage) and micro (melanosome morphology) autapomorphies. It is a remarkable specimen!

I definitely appreciate the authors' undertaking of collecting much data and conducting statistical analyses, however, those are only robust if the a priori assumptions are accurate. The authors argue that their assumptions are valid. However, I agree with Reviewer 2 and disagree with the authors. It is not appropriate to make these sweeping assumptions for reasons below.

Lines 265-269: Indeed, over ~160 million years this specimen was subjected to many taphonomic and diagenetic processes that likely altered the original morphology of the feathers, as demonstrated by the authors' observation that the feathers are 'too densely preserved to display gross and fine morphological features'. The obvious concern then is how can the authors use the arrangement and measurement of microstructures to make specific and accurate interpretations of plumage coloration, based on comparative analyses with unaltered modern samples, if these fossil structures have been altered, both macro and microscopically?? In other words, the authors are drawing their conclusions based on comparisons between fossil and modern microstructures that are not directly comparable because the fossils have been altered but the modern has not. The alteration of the fossil feathers during preservation is one of the many reasons scientists cannot rely only on morphology alone to definitively identify the origin of the microstructures as melanosomes, let alone interpret feather coloration.

Lines 380-387: I appreciate the transparency of this disclaimer. However, over the past several years, criteria have been proposed in several publications (cited by the authors and Reviewer 2) for testing and identifying the fossil feather microstructures because of the morphological similarities between fossil bacteria and melanosomes.

It is not appropriate to assume a melanosome origin just because a couple of the recent studies on fossil feathers, which employ various additional techniques, have resulted in data supporting the microstructures as melanosomes. The assignment of a melanosome origin for the microbodies is a direct result of the data generated from using appropriate techniques (TEM, Tof-SIMS, IHC), for each sample in each study. In addition, a taphonomic study on modern snake skin (Schweitzer et al. 2015) and a study of fossil feathers (Lindgren et al. 2015) have shown that melanosomes and bacteria cells may both be present, even in the same microscopic focal field. In summary, it has been demonstrated that 1) bacteria cells and melanosomes are indistinguishable using only SEM because their size, morphologies and arrangement overlap, 2) both can preserve, 3) both can be present in the same microscopic field (modern and fossil) and 4) coloration is much more complex than presence of melanosomes. Therefore, contrary to the authors' response, it is necessary to spend the time and resources to test each sample and fossil with molecular techniques before ascribing a melanosome origin to all microstructures!

As the authors and reviewer have noted, not only has caution been expressed in the scientific literature regarding the identification of fossil melanosomes, but even more so for the interpretation of color of extinct animals. Two of the authors (MDS and LD) have published a couple recent articles that directly address some of the factors affecting feather degradation and optical properties of feathers. The one study looks at how melanized feathers are degraded compared to unmelanized (white) feathers (Justyn et al. 2017). One is a review on the coloration of animal integuments which echoes the theme that coloration and other perceived characteristics of feathers are complex and can be a transient feature based on the animal's lifestyle, molt, etc. (Shawkey and D'Alba 2017). In light of the conclusions and implications expressed in these recent papers, the authors need to explain how degradation is accounted for in this specimen (i.e. are they only observing melanosome morphologies indicative of black feathers because the unmelanized feathers or regions are degraded?) and how they account for other pigments and/or optical properties that may have differing preservation potentials (i.e. what about structural coloration? Is keratin preserved? If so, is it unaltered? How does this affect interpretation of color or appearance?).

Line 415: I don't think it's appropriate to use the term 'platelets' as the noun replacement for melanosomes because platelets are a specific biological structure. I think 'platelet-shaped melanosomes' must always be used.

Line 423: I agree with the reviewer that FIB is problematic and does not provide an accurate cross-section through the structure like TEM generates, which is why TEM has been proposed as one of required pieces of data for determining the origin of the microstructures.

References

- Justyn, N. M., Peteya, J. A., D'Alba, L., & Shawkey, M. D. (2017). Preferential attachment and colonization of the keratinolytic bacterium *Bacillus licheniformis* on black- and white-striped feathers. *The Auk*, *134*(2), 466–473. <http://doi.org/10.1642/AUK-16-245.1>
- Lindgren, J., Sjövall, P., Carney, R. M., Cincotta, A., Uvdal, P., Hutcheson, S. W., ... Godefroit, P. (2015). Molecular composition and ultrastructure of Jurassic paravian feathers. *Scientific Reports*, *5*, 13520. <http://doi.org/10.1038/srep13520>
- Schweitzer, M. H., Lindgren, J., & Moyer, A. E. (2015). Melanosomes and ancient coloration re-examined: A response to Vinther 2015 (DOI 10.1002/bies.201500018). *BioEssays*, *37*(11), 1174–1183. <http://doi.org/10.1002/bies.201500061>
- Shawkey, M. D., & D'Alba, L. (2017). Interactions between colour-producing mechanisms and their effects on the integumentary colour palette. *Philosophical Transactions of the Royal Society of London B: Biological Sciences*, *372*(1724). Retrieved from <http://rstb.royalsocietypublishing.org/content/372/1724/20160536>